# Factor XII signaling via uPAR-integrin β1 axis promotes tubular senescence in diabetic kidney disease

Ahmed Elwakiel [1] ✉, Dheerendra Gupta[1], Rajiv Rana[1], Jayakumar Manoharan[1], Moh'd Mohanad Al-Dabet[1,12], Saira Ambreen[1], Sameen Fatima[1], Silke Zimmermann[1], Akash Mathew[1], Zhiyang Li[1], Kunal Singh[1], Anubhuti Gupta[1], Surinder Pal[1], Alba Sulaj [2], Stefan Kopf[2], Constantin Schwab[3], Ronny Baber[1,4], Robert Geffers [5], Tom Götze[6], Bekas Alo[6], Christina Lamers[6], Paul Kluge[6], Georg Kuenze[6,7], Shrey Kohli [1], Thomas Renné[8,9,10], Khurrum Shahzad[1,11] & Berend Isermann [1] ✉

Coagulation factor XII (FXII) conveys various functions as an active protease that promotes thrombosis and inflammation, and as a zymogen via surface receptors like urokinase-type plasminogen activator receptor (uPAR). While plasma levels of FXII are increased in diabetes mellitus and diabetic kidney disease (DKD), a pathogenic role of FXII in DKD remains unknown. Here we show that FXII is locally expressed in kidney tubular cells and that urinary FXII correlates with kidney dysfunction in DKD patients. *F12*-deficient mice (*F12⁻/⁻*) are protected from hyperglycemia-induced kidney injury. Mechanistically, FXII interacts with uPAR on tubular cells promoting integrin β1-dependent signaling. This signaling axis induces oxidative stress, persistent DNA damage and senescence. Blocking uPAR or integrin β1 ameliorates FXII-induced tubular cell injury. Our findings demonstrate that FXII-uPAR-integrin β1 signaling on tubular cells drives senescence. These findings imply previously undescribed diagnostic and therapeutic approaches to detect or treat DKD and possibly other senescence-associated diseases.

Coagulation factor XII (FXII, gene *F12*) is activated upon interaction with negatively charged surfaces (contact activation). The activated protease (FXIIa) initiates the intrinsic coagulation pathway and inflammatory reactions via the kallikrein-kinin-system (KKS)[1].

Furthermore, FXII zymogen signals through plasma membrane receptors such as the urokinase-type plasminogen activator receptor (uPAR) in different cells promoting cell and context- specific responses, including angiogenic effects in endothelial cells, activation of

[1]Institute of Laboratory Medicine, Clinical Chemistry and Molecular Diagnostics, University of Leipzig Medical Center, Leipzig, Germany. [2]Internal Medicine I and Clinical Chemistry, German Diabetes Center (DZD), University of Heidelberg, Heidelberg, Germany. [3]Institute of pathology, University of Heidelberg, Heidelberg, Germany. [4]Leipzig Medical Biobank, Leipzig University, Leipzig, Germany. [5]Genome Analytics, Helmholtz Centre for Infection Research, Braunschweig, Germany. [6]Institute for Drug Discovery, Faculty of Medicine, Leipzig University, Leipzig, Germany. [7]Center for Scalable Data Analytics and Artificial Intelligence, Leipzig University, Leipzig, Germany. [8]Institute of Clinical Chemistry and Laboratory Medicine, University Medical Center Hamburg-Eppendorf, Hamburg, Germany. [9]Center for Thrombosis and Hemostasis (CTH), Johannes Gutenberg University Medical Center, Mainz, Germany. [10]Irish Centre for Vascular Biology, School of Pharmacy and Biomolecular Sciences, Royal College of Surgeons in Ireland, Dublin, Ireland. [11]National Centre of Excellence in Molecular Biology, University of the Punjab, 87-West Canal Bank Road, Lahore, Pakistan. [12]Present address: Department of Medical Laboratory Sciences, School of Science, University of Jordan, Amman, Jordan. ✉e-mail: ahmed.elwakiel@medizin.uni-leipzig.de; berend.isermann@medizin.uni-leipzig.de

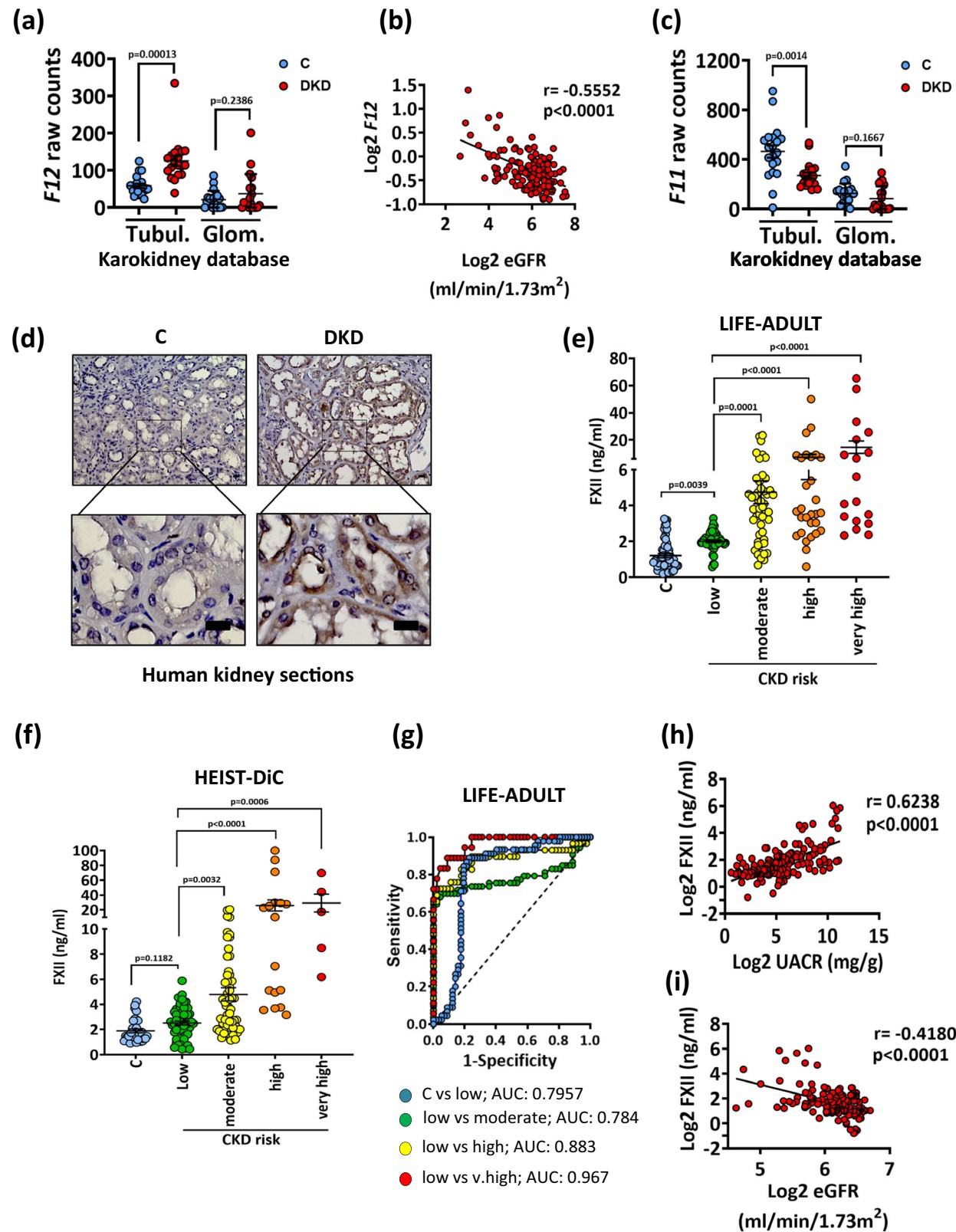

immune cells such as neutrophils and macrophages, and profibrotic effects in lung fibroblasts[2–6].

Proinflammatory and profibrotic signaling is a hallmark of diabetic kidney disease (DKD), a serious microvascular complication in patients with diabetes mellitus[7,8]. The pathomechanisms underlying proinflammatory and profibrotic signaling in DKD involve hemodynamic and metabolic changes as well as DNA damage and

senescence[9,10]. Cellular stressors, such as increased reactive oxygen species (ROS) generation in diabetic kidneys, trigger DNA damage and senescence, which is characterized by permanent cell cycle arrest, macromolecular damage, morphological changes, and a specific secretome (SASP, senescence-associated secretory phenotype) that induces inflammatory and fibrotic changes and compromises kidney function[10,11].

**Fig. 1 | Upregulation of FXII correlates with impaired kidney function in human DKD. a** Dot-plot summarizing *F12* expression in the tubulointerstitial and the glomerular compartments (Karokidney public RNA-sequencing database). Dot-plots reflecting mean ± SEM of 20 controls (C) and 19 DKD samples; two-tailed unpaired student's *t* test. **b** Line-graph representing the negative correlation of *F12* expression in the tubulointerstitium with the estimated glomerular filtration rate (eGFR) in CKD patients and in healthy living donors (*n* = 147) from the Ju CKD TubInt Dataset of the Nephroseq® database. The confidence interval of r (Pearson coefficient) and *P* value (two-tailed) were calculated by linear regression. **c** Dot-plot summarizing *F11* expression in the tubulointerstitial and the glomerular compartments (Karokidney public RNA-sequencing database). Dot-plots reflecting mean ± SEM of 20 controls (C) and 19 DKD samples; two-tailed unpaired student's *t* test. **d** Exemplary histological images of human kidney sections stained for FXII (top) and magnified areas (bottom) obtained from nondiabetic controls (C; *n* = 6) or diabetic patients with DKD (DKD; *n* = 5). Scale bars represent 20 μm. **e, f** Dot-plots showing the distribution of the urinary levels of FXII (ng/ml; ELISA) in urine samples

obtained from the LIFE-ADULT (**e**) and HEIST-DiC (**f**) cohorts (number of samples are provided in Supplementary Tables S2 and S3 respectively). Urinary FXII was measured in normoglycemic controls (C) and in diabetic individuals (CKD grade according to KDIGO criteria). Dot-plots reflecting mean ± SEM; Kruskal-Wallis test with Dunn's multiple comparison test. **g** Receiver operating characteristic (ROC) analyses of urinary FXII (ng/ml; ELISA) in diabetic individuals with low risk of CKD compared to nondiabetic controls (blue) or in diabetic individuals with moderate risk of CKD (green), high risk (yellow), and very high risk (red) compared to low-risk patients in the LIFE-ADULT cohort. AUC: area under the curve. **h, i** Line graphs representing the positive correlation of urinary FXII (ng/ml) with urinary albumin creatinine ration (UACR; mg albumin/g creatinine; (**h**) *n* = 140) and the negative correlation with the estimated glomerular filtration rate (eGFR, ml/min/1.73 m²; (**i**) *n* = 138) in diabetic individuals from the LIFE-ADULT cohort. The confidence interval of r (Pearson coefficient) and *P* values (two-tailed) were calculated by linear regression. Source data are provided as a "Source Data" file.

In addition, diabetes mellitus, in general, and DKD, in particular, are linked with alterations of the coagulation system that predispose to inflammatory and fibrotic changes[12,13]. Despite its known role in thrombosis and bradykinin-driven inflammation, the role of FXII in the pathophysiology of DKD is not yet defined. Earlier reports showed upregulation of hepatic FXII production in patients with insulin resistance and increased levels of circulating FXII and FXIIa in patients with diabetes mellitus or chronic kidney disease (CKD)[14–17]. Neutrophil-derived FXII activates neutrophils in an autocrine fashion, demonstrating that non-hepatic FXII promotes inflammation[4], a key feature of DKD.

FXII signaling via uPAR involves uPAR coreceptors, such as integrins, promoting cell-specific responses. Interference with FXII-uPAR binding can inhibit FXII-associated signaling effects[3,4]. uPAR is associated with senescence, and targeting uPAR with CAR-T cells eliminates senescent cells and associated pathologies[18]. Furthermore, the role of uPAR in renal diseases, including DKD, is established[19,20].

Whether FXII induction in DKD contributes to the inflammatory state through the activation of the intrinsic coagulation pathway, the KKS, or is mechanistically linked with DKD through a signaling mechanism independent of its protease function remains unknown. It is possible, but it remains to be shown that the interaction of FXII with uPAR promotes senescence in diabetic kidneys. Deciphering the relevance of FXII binding to uPAR for induction of senescence may be therapeutically relevant, as strategies inhibiting zymogen FXII or its active form do not increase the risk of bleeding and are hence considered safe[21].

In the current study, we combined unbiased approaches, analyses of diabetes patient samples, and murine diabetes models to uncover a function of zymogen FXII signaling for DKD pathology, which promotes oxidative DNA damage and tubular senescence via uPAR-integrin β1 signaling.

## Results

### Kidney FXII induction correlates with impaired function in human DKD

Transcriptomic analysis of the Nephroseq® and the Karolinska kidney research center (Karokidney[22]) databases revealed increased kidney tubular *F12* expression in DKD patients, while the glomerular expression remained unchanged (Fig. 1a and Supplementary Fig. S1a, b). *F12* expression was inversely correlated with the estimated glomerular filtration rate (eGFR; Fig. 1b). In contrast, tubular expression of *F11*, which codes for coagulation factor XI, the FXIIa substrate in the intrinsic pathway, was suppressed in DKD as compared to nondiabetic kidneys (Fig. 1c and Supplementary Fig. S1a,b).

To investigate whether increased *F12* expression translates into increased protein expression, we next analyzed FXII protein

expression in human kidney biopsies of diabetic patients with established DKD. DKD in these patients was characterized by tubulointerstitial fibrosis, thickening of the tubular basement membrane, and glomerular mesangial matrix expansion (Supplementary Fig. S1c and Supplementary Table S1). FXII was markedly increased in the tubular compartment in DKD biopsies compared to nondiabetic controls (Fig. 1d).

Considering that FXII is a secreted protein, we investigated whether urinary FXII levels were increased in patients with DKD in two independent large cross-sectional cohorts of type-2 diabetic patients with different stages of CKD (LIFE-ADULT; Supplementary Table S2 and HEIST-DiC, Supplementary Table S3). Urinary FXII levels were increased in DKD patients compared to healthy controls and were associated with DKD severity in both cohorts (ELISA; Fig. 1e, f). Receiver operating characteristic (ROC) curve analyses revealed an area under the curve (AUC) of 0.7957 for urinary FXII levels in diabetic patients with a low CKD risk compared to healthy controls, and the AUC was even higher when patients with a higher CKD risk were compared to those with low risk in the LIFE-ADULT cohort (Fig. 1g). Similarly, urinary FXII levels were associated with CKD severity in the HEIST-DiC cohort (SupplementaryFig. S1d). Concomitantly, urinary FXII levels positively correlated with albuminuria and cystatin C levels (Fig. 1h and Supplementary Fig. S1e, f) and negatively with eGFR (Fig. 1i and Supplementary Fig. S1g) in both cohorts. Taken together, kidney FXII expression is induced in DKD patients, and increased urinary FXII levels reflect impaired kidney function.

### *F12⁻/⁻* mice have reduced kidney dysfunction and histopathological changes in experimental DKD

To investigate the role of FXII in DKD, we induced persistent hyperglycemia in wild-type (WT) and FXII deficient mice (*F12⁻/⁻*) mice using streptozotocin (STZ) for 24 weeks (Fig. 2a). Similar to the findings in humans, FXII expression (mRNA and protein) was upregulated predominantly in the tubular compartment in the kidneys of hyperglycemic WT mice compared to normoglycemic controls (Fig. 2b and Supplementary S2a). In db/db mice (representing type-2 diabetes[23]), FXII expression (protein and mRNA) was likewise increased in the tubular compartment compared to nondiabetic control db/m mice (Supplementary Fig. S2b, c).

Kidney functional and histopathological markers were indistinguishable between normoglycemic WT and *F12⁻/⁻* mice (Fig. 2c–g and Supplementary Fig. S2d–k). While blood glucose level and body weight were comparable in hyperglycemic WT and *F12⁻/⁻* mice after 24 weeks of persistent hyperglycemia (Supplementary Fig. S2d, e), the latter group displayed improved kidney function, as indicated by reductions in the urinary albumin/creatinine ratio (UACR), blood urea nitrogen (BUN), and normalized kidney weight (Fig. 2c–e). Histological analyses of hyperglycemic *F12⁻/⁻* kidneys revealed reduced DKD-associated

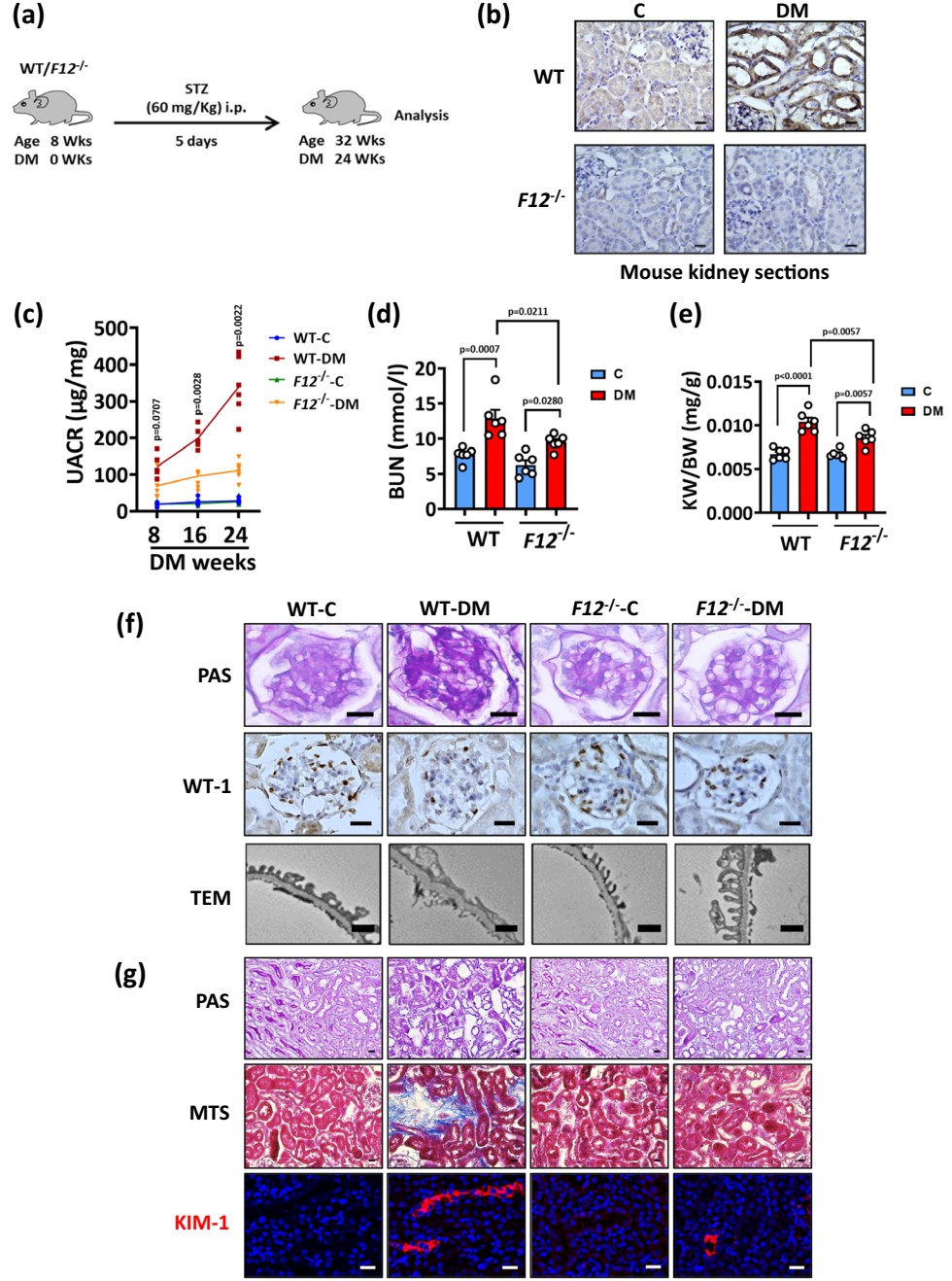

**Fig. 2 | *F12^-/-* mice are protected from DKD. a** Experimental scheme of the DKD model. Wild type (WT) and *F12^-/-* mice were age-matched, and persistent hyperglycemia was induced using streptozotocin (STZ) for 24 weeks. **b** Exemplary histological images of kidney sections stained for FXII comparing normoglycemic controls (C) and hyperglycemic (DM) WT and *F12^-/-* mice. FXII is detected by HRP-DAB reaction (brown); hematoxylin nuclear counter stain (blue). Scale bars represent 20 μm. **c** Line graphs showing urinary albumin-creatinine ratio (UACR, μg albumin/mg creatinine) in experimental groups (as described in **b**) after 8, 16, or 24 weeks of persistent hyperglycemia. Line graphs reflecting mean ± SEM of 6 mice per group; two-way ANOVA with Tukeys's multiple comparison test comparing hyperglycemic WT and *F12^-/-* mice at the 3 time points. **d, e** Dot-plots summarizing blood urea nitrogen (BUN, mmol/l; (**d**) and adjusted kidney weight (KW/BW, mg kidney weight /g body weight; (**e**) in the experimental groups (as described in **b**).

Dot-plots reflecting mean ± SEM of 6 mice per group; two-way ANOVA with Tukeys's multiple comparison test. **f** Exemplary histological images of periodic acid Schiff staining (PAS) showing glomeruli (top panel), podocyte number reflected by Wilms tumor 1 immunostaining (WT-1, brown, hematoxylin nuclear counterstain, blue; middle panel), and transmission electron microscopy of podocytes (TEM; bottom panel) in experimental groups (as described in **b**); scale bars of top and middle panels represent 20 μm, while scale bars of bottom panel represent 1 μm. **g** Exemplary histological images of periodic acid Schiff staining (PAS) showing tubuli (top panel), interstitial fibrosis (middle panel, Masson's trichrome stain, MTS), and kidney injury molecule-1 immunostaining (bottom panel, KIM-1, red; DAPI nuclear counterstain, blue) in experimental groups (as described in **b**); all scale bars represent 20 μm. Source data are provided as a "Source Data" file.

glomerular lesions, including mesangial matrix expansion, loss of podocytes, and thickening of the glomerular basement membrane (GMB) (Fig. 2f and Supplementary Fig. S2f–h). Furthermore, tubular pathology (tubular dilation, atrophy, and loss of brush borders),

tubulointerstitial fibrosis, and the expression of kidney injury molecule-1 (KIM-1) appeared less pronounced in hyperglycemic *F12^-/-* mice (Fig. 2g and Supplementary Fig. S1i, k). Thus, FXII deficiency ameliorates glomerular and tubular damage in experimental murine DKD.

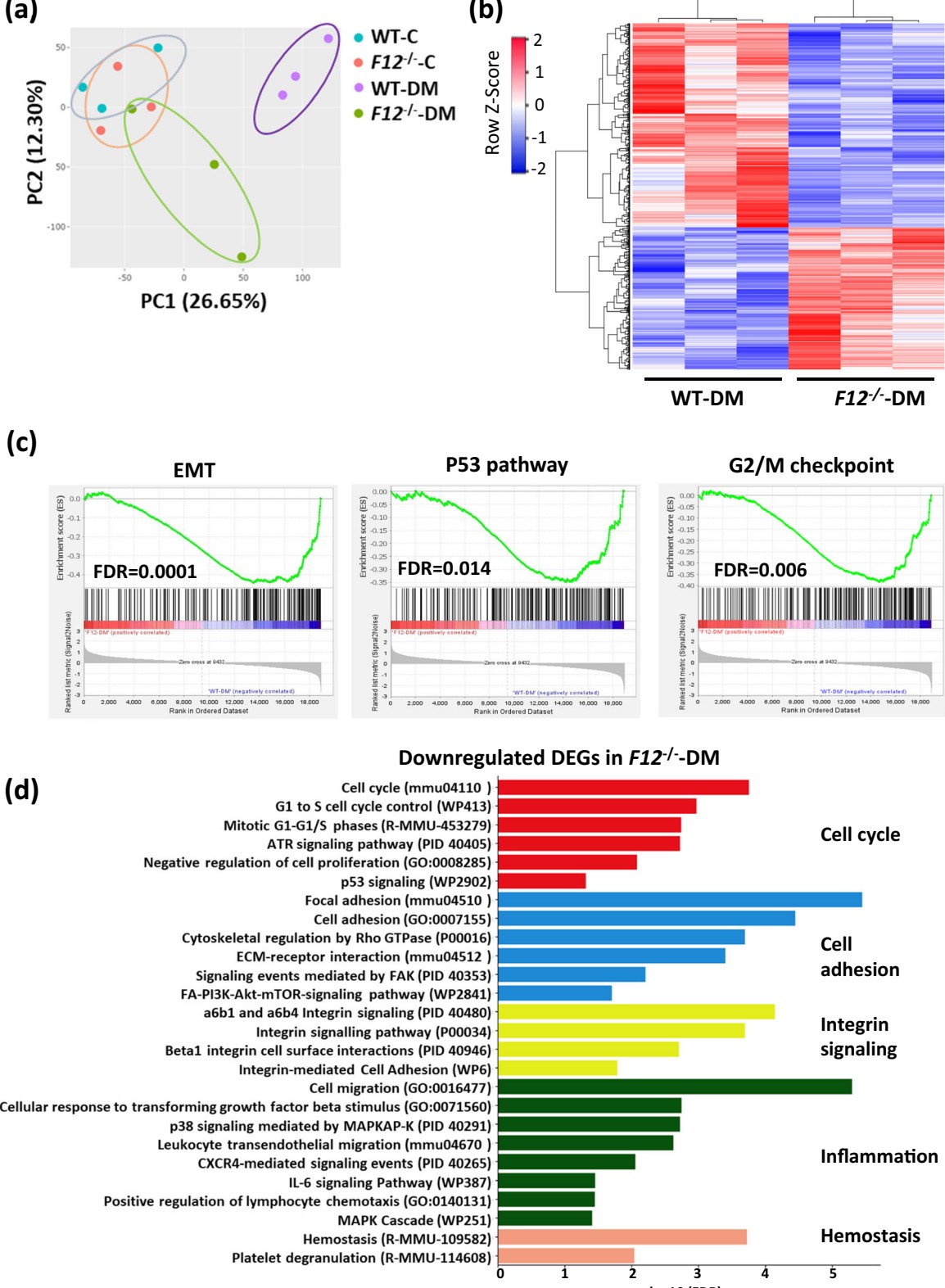

**Fig. 3 | FXII deficiency induces differential gene expression in DKD. a** Principal component analysis (PCA) on gene sets of normoglycemic (C) and hyperglycemic (DM) WT and *F12*[-/-] mice kidneys. **b** Heatmap of the RNA-seq data showing the differentially expressed genes (DEGs) in WT-DM and *F12*[-/-]-DM mice. Each column represents data from an individual mouse. Color intensity represents row Z-score. **c** Gene set enrichment analysis (GSEA) plots of the hallmark gene sets representing key negatively enriched pathways when comparing *F12*[-/-]-DM to WT-DM kidneys. Significance is represented by the false discovery rate (FDR). **d** Bar graph representing the top enriched pathways based on the downregulated differentially expressed genes (DEGs) in *F12*[-/-]-DM compared to WT-DM kidneys using KEGG (Kyoto Encyclopedia of Genes and Genomes), WikiPathways, Reactome, PID (Pathway Interaction Database), and GO (Gene Ontology: Biological processes) databases. The pathways were ranked by the false discovery rate (FDR). Source data are provided as a "Source Data" file.

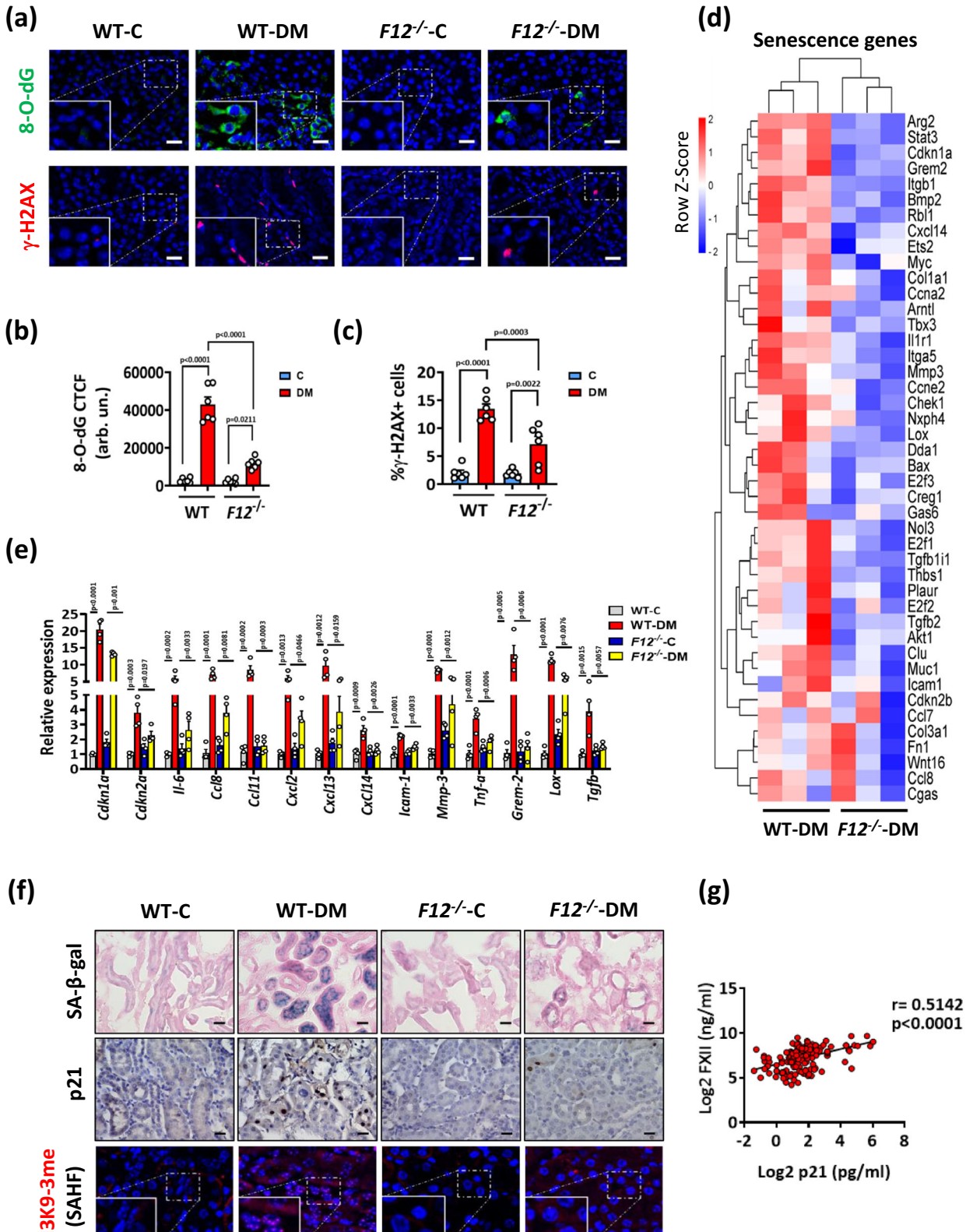

## FXII deficiency induces differential gene expression in DKD

To gain mechanistic insights on FXII roles in DKD, we performed bulk RNA sequencing (RNA-seq) on the kidneys of WT and $F12^{-/-}$ mice. Expression patterns of normoglycemic WT and $F12^{-/-}$ mice were similar, while the expression signatures of hyperglycemic WT and $F12^{-/-}$ mice differed, as revealed by principal component analysis (PCA) (Fig. 3a). The clustering of mRNA indicated differential gene expression with 614 genes downregulated and 428 genes upregulated in hyperglycemic $F12^{-/-}$ compared to WT mice (Fig. 3b). Gene set enrichment analysis (GSEA) using the hallmark gene sets revealed a negative enrichment of fibrosis-related pathways (epithelial-mesenchymal transition; EMT), cell cycle arrest (p53 pathway and G2/M checkpoints), stress pathways (mTORC1 and the unfolded protein response), and inflammation (IL2-STAT5 signaling) in hyperglycemic

**Fig. 4 | *F12*[-/-] mice kidneys are less susceptible to DNA damage and associated senescence. a–c** Exemplary images (**a**) and dot blots summarizing results (**b, c**) of 8-hydroxy-2'-deoxyguanosine (8-O-dG) and phosphorylated H2A histone X (γ-H2AX) staining comparing normoglycemic (control, C) and hyperglycemic (DM) wild type (WT) and *F12*[-/-] mice. 8-O-dG and γ-H2AX are immunofluorescently detected, green and red, respectively; DAPI nuclear counterstain, blue. Insets show higher magnification of the marked areas. Scale bars represent 20 μm. Dot-plots reflecting mean ± SEM of 6 mice per group; two-way ANOVA with Tukeys's multiple comparison test. CTCF: corrected total cell fluorescence. Arb. un.: arbitrary units. **d** Heatmap of the RNA-seq data showing gene expression changes of senescence-associated genes in WT-DM and *F12*[-/-]-DM mice. Each column represents data from an individual mouse. Color intensity represents row Z-score. **e** Bar graphs summarizing expression (qRT-PCR) of selected senescence-associated genes in

experimental groups (as described in **a**). Bar graphs reflecting mean ± SEM of 4 mice per group; two-way ANOVA with Tukeys's multiple comparison test. **f** Exemplary images of mouse kidney sections stained for senescence-associated β-galactosidase (top panel, SA-β-gal, blue; eosin counterstain), p21 (middle panel, detected by HRP-DAB reaction, brown; hematoxylin nuclear counter stain, blue), and senescence-associated heterochromatin foci of tri-methyl-histone H3 (Lys9) (bottom panel, H3K9-3me SAHF, immunofluorescently detected, red; DAPI nuclear counterstain, blue, insets show higher magnification of the marked areas) in experimental groups (as described in a). Scale bars represent 20 μm. **g** Line graph representing the positive correlation of urinary FXII (ng/ml) with urinary p21 (pg/ml) in diabetic individuals from the LIFE-ADULT cohort (*n* = 146). The confidence interval of r (Pearson coefficient) and *P* values (two-tailed) were calculated by linear regression. Source data are provided as a "Source Data" file.

*F12*[-/-] mice (Fig. 3c and Supplementary Fig. S3a). Further pathway analysis of the differentially expressed genes (DEGs) indicated that FXII deficiency was associated with the downregulation of pathways involved in cell cycle regulation, cell adhesion, integrin signaling, inflammation, and hemostasis (Fig. 3d and Supplementary Fig. S3b-d). On the other hand, compared to WT mice, *F12* deficiency (i) upregulated DNA damage repair pathways and (ii) upregulated metabolic pathways such as lipid metabolism, amino acid metabolism, and organic acid metabolism[24,25] (Supplementary Fig. S4). Collectively, FXII regulates gene sets related to pathways linked to DKD in hyperglycemic kidneys[12,26–28].

### Hyperglycemic *F12*[-/-] mice are less susceptible to kidney DNA damage-associated senescence

DNA damage activates cell cycle arrest and cell cycle checkpoints to prevent genomic instability[29,30]. The negative enrichment of pathways related to cell cycle arrest (p53 pathway) and cell cycle checkpoints (G1/S and G2/M checkpoints) in hyperglycemic *F12*[-/-] mice (Fig. 3d and Supplementary Fig. S4a-c) prompted us to analyze the extent of reactive oxygen species (ROS)-induced DNA damage. Kidney sections were stained for 8-hydroxy-2'-deoxyguanosine (8-O-dG), a marker of oxidative DNA damage[31], and phosphorylated H2A histone X (γ-H2AX), a marker of unrepaired DNA damage[32]. ROS-induced DNA damage was reduced in hyperglycemic *F12*[-/-] mice (Fig. 4a–c), suggesting that FXII promotes ROS generation while impairing the DNA damage response in DKD.

Persistent DNA damage and defective repair during DKD trigger premature senescence that further promotes DKD progression[33]. The cell cycle inhibitor p21 (*Cdkn1a*), which we have identified as a driver of tubular cell senescence in DKD[34], was downregulated in hyperglycemic *F12*[-/-] compared to WT mice (Supplementary Fig. S3b and Supplementary Fig. S5a-c). Furthermore, we analyzed a panel of genes known to be associated with senescence in mice[35,36]. These genes were downregulated in the kidneys of hyperglycemic *F12*[-/-] compared to WT mice (Fig. 4d). The downregulation of genes related to cell cycle arrest, the senescence-associated secretory phenotype (SASP), cell adhesion, and fibrosis in the kidneys of hyperglycemic *F12*[-/-] mice was confirmed by qRT-PCR (Fig. 4e). In addition, analysis of selected SASP-related cytokines and chemokines using Olink technology revealed increased SASP-related cytokines and chemokines in some hyperglycemic WT, but not in hyperglycemic *F12*[-/-] mice kidneys (Supplementary Fig. S5d). Reduction of tubular senescence in the kidneys of hyperglycemic *F12*[-/-] mice was verified by immunostaining for the senescence-associated β-galactosidase (SA-β-gal), p21, the proliferation marker Ki-67, the nuclear envelope protein lamin-B1, and senescence-associated heterochromatin foci (SAHF) (Fig. 4f and Supplementary Fig. S6a–f). In humans, urinary FXII was positively correlated with urinary p21 in DKD patients of the LIFE-ADULT cohort (Fig. 4g).

SASP chemokines promote the infiltration of inflammatory cells[37], and hence we determined macrophage proportions using F4/80 immunostaining. We observed reduced macrophage infiltration

in the tubulointerstitium of hyperglycemic *F12*[-/-] mice compared to hyperglycemic WT mice (Supplementary Fig. S6g, h). Senescent cells induce anti-apoptotic regulators and are not hallmarked by increased apoptosis[38,39]. The anti-apoptotic regulators BCL-2 and BCL-XL were induced in hyperglycemic WT compared to *F12*[-/-] mice kidneys, and the number of cleaved caspase-3 positive cells was not different in the kidneys of both genotypes (Supplementary Fig. S7). These findings are consistent with increased kidney senescence in WT, but not *F12*[-/-] mice. Thus, loss of FXII expression protects mice from senescence and inflammation in DKD.

### FXII is predominantly expressed by murine and human renal tubular cells

Analysis of the Nephroseq® database confirmed higher *F12* expression in the tubulointerstitial compartment compared to the glomerular compartment (Supplementary Fig. S8a), consistent with our immunostaining data in human and mouse kidneys. Furthermore, analysis of *F12* expression in a single cell transcriptomic database (Kidney Interactive Transcriptomics; KIT) indicated the highest *F12* expression in clusters of proximal tubular cells in healthy adult kidneys[40] (Supplementary Fig. S8b). In addition, *F12* expression was induced in proximal tubular clusters of DKD patients[41,42] (Supplementary Fig. S8c, d). FXII expression was readily detectable in a human proximal tubular cell line (HKC-8) and in murine primary proximal tubular cells (PTCs) and was increased following stimulation with high glucose (25 mM, 24 h) (Supplementary Fig. S8e–h). Thus, FXII is predominantly upregulated in the kidney tubular compartment under hyperglycemic conditions, supporting a model in which the induction of FXII in tubular cells promotes senescence in DKD.

### FXII induces DNA damage and associated senescence in kidney tubular cells in vitro

Considering the zymogen FXII's direct cellular effects beyond coagulation and KKS activation[43], we next examined whether FXII can directly promote DNA damage and premature senescence in kidney tubular cells. Markers of DNA damage and senescence were induced in HKC-8 cells exposed to increasing concentrations of purified human FXII for 24 h in the presence of Zn²⁺, a cofactor for FXII cell surface binding[2] (Supplementary Fig. S9a, b). FXII (62 nM) induced time-dependently the expression of KIM-1, p21, and γ-H2AX in HKC-8 cells starting from 6 h (Fig. 5a, b). Mouse PTCs exposed to a similar dose of recombinant murine FXII showed comparable induction of these markers (Supplementary Fig. S9c, d). The induction of tubular cell injury and senescence by FXII was paralleled by increases in intracellular ROS levels (H2DCFDA) and oxidative DNA damage (8-O-dG) in HKC-8 cells and in PTCs (Fig. 5c–e and Supplementary Fig. S9e, f). Cell cycle analysis of PTCs exposed to FXII revealed an increase of cells accumulating in the G2/M phase compared to untreated cells (Supplementary Fig. S9g, h). Furthermore, FXII induced SA-β-gal signal in PTCs and the expression of genes related to cell cycle arrest, SASP, and inflammation in HKC-8 cells (Fig. 5f–h). Thus, FXII induces ROS

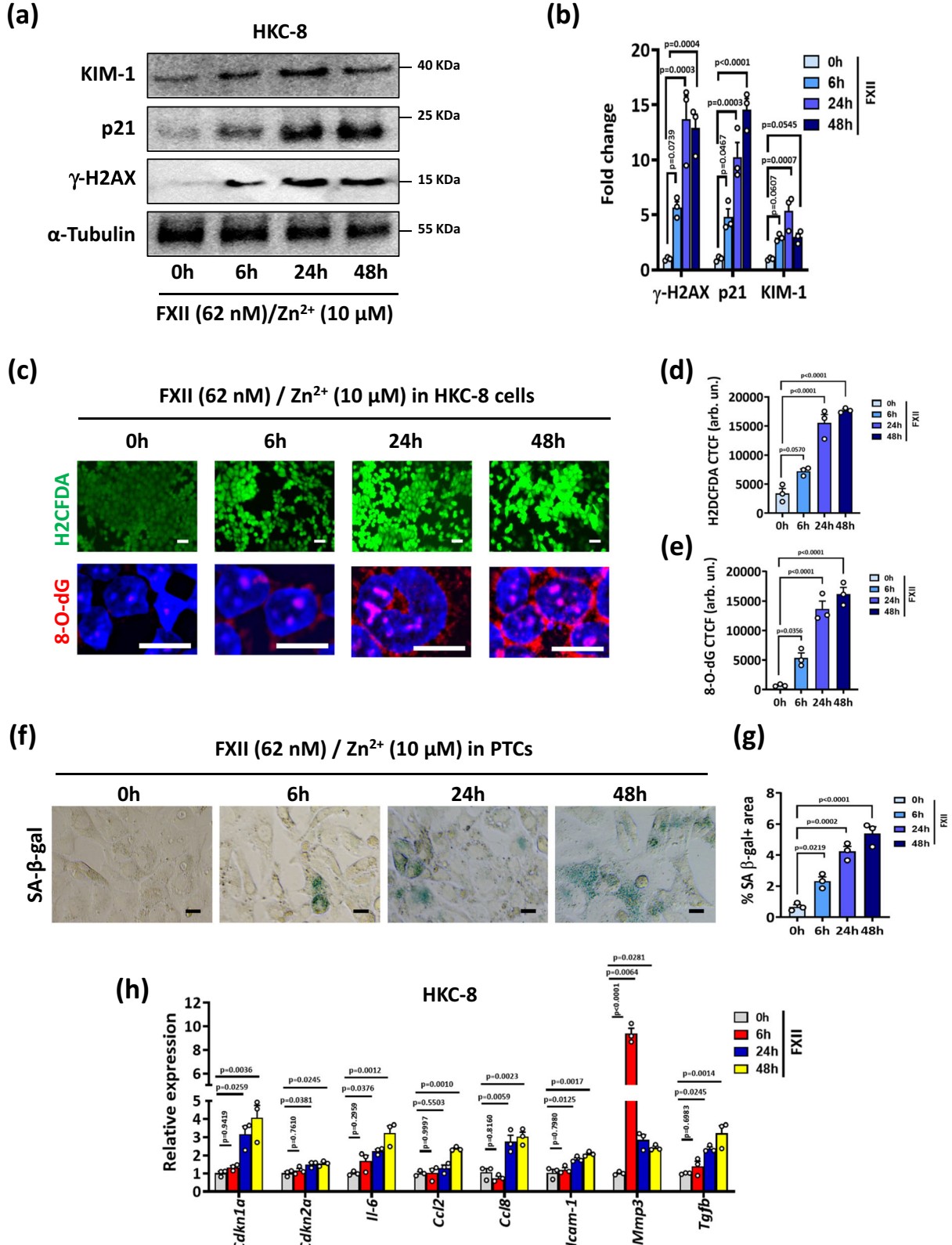

accumulation, DNA damage and senescence in human and murine kidney tubular cells in vitro.

## FXII interacts with uPAR on tubular cells under hyperglycemic conditions

To address a possible role of uPAR in FXII-mediated tubular senescence, we analyzed its expression in hyperglycemic WT and *F12^-/-*

kidneys. uPAR induction was strong in WT kidneys, however, the receptor was hardly detectable in *F12^-/-* mouse kidneys (Fig. 6a, b and Supplementary Fig. S10a). Exposure of HKC-8 cells to human FXII increased the surface expression of uPAR compared to control cells (Fig. 6c,d and Supplementary Fig. S10b). To investigate whether FXII binds to uPAR on the tubular cell surface with glucose stimulation, we exposed PTCs and HKC-8 cells to high glucose (25 mM, 24 h) and

**Fig. 5 | FXII induces DNA damage and associated senescence in kidney tubular cells in vitro. a, b** Representative immunoblots (**a** loading control: α-Tubulin) and dot-plots summarizing results (**b**) for γ-H2AX, p21, and KIM-1 expression in HKC-8 cells exposed to purified human FXII (62 nM) in the presence of $Zn^{2+}$ (10 μM) for 6, 24, and 48 h. Dot-plots reflecting mean ± SEM of 3 independent experiments; one-way ANOVA with Tukeys's multiple comparison test. **c–e** Exemplary images (**c**) and dot-plots summarizing results (**d, e**) of staining with the intracellular ROS detector 2′,7′-dichlorodihydrofluorescein diacetate (top panel, H2DCFDA, green) and 8-hydroxy-2′-deoxyguanosine (bottom panel, 8-O-dG, red; DAPI nuclear counterstain, blue) in experimental groups (as described in **a**). Scale bars represent 20 μm. Dot-plots reflecting mean ± SEM of 3 independent experiments; one-way ANOVA with

Tukeys's multiple comparison test. CTCF corrected total cell fluorescence. Arb. un.: arbitrary units. **f, g** Exemplary images of senescence-associated β-galactosidase (**f**, SA-β-gal, blue) and dot-plot summarizing quantification of SA-β-gal staining (**g**) in PTCs treated with recombinant mouse FXII (62 nM) in the presence of $Zn^{2+}$ (10 μM) for 6, 24, and 48 h. Dot-plots reflecting mean ± SEM of 3 independent experiments; one-way ANOVA with Tukeys's multiple comparison test. **h** Bar graphs summarizing expression (qRT-PCR) of selected senescence-associated genes in experimental groups (as described in **a**). Bar graphs reflecting mean ± SEM of 3 independent experiments; one-way ANOVA with Tukeys's multiple comparison test. Source data are provided as a "Source Data" file.

determined FXII-uPAR interaction by proximity ligation assay (PLA) and co-immunoprecipitation. Both assays revealed an increase in the FXII-uPAR interaction in high glucose conditions (Fig. 6e, f and Supplementary Fig. S10c). Furthermore, increased interaction of FXII and uPAR was detected in human DKD biopsies compared to control biopsies using PLA (Fig. 6g, h). To investigate whether other plasma proteins that compete with FXII to uPAR binding such as high molecular weight kininogen (HK) and vitronectin[2] may interfere with FXII-uPAR binding, we analyzed the Nephroseq® and the Karokidney transcriptomic databases. While tubular *F12* expression was increased in DKD, the expression of *KNG1* (encoding HK) was downregulated, and *VTN* (encoding vitronectin) was not changed compared to controls (Supplementary Fig. S11a, b). Analysis of FXII, HK, and uPAR expression in human DKD biopsies revealed upregulation and colocalization of FXII and uPAR, while the HK signal was barely detectable and showed little if any colocalization with the upregulated uPAR (Supplementary Fig. S11c g). To experimentally address the effects of competing proteins on FXII-uPAR binding on renal tubular cells, we exposed HKC-8 cells to FXII (62 nM; ±10 μM zinc) in the presence of equimolar (62 nM) or excess molar (120 nM) concentrations of HK and determined FXII binding to uPAR by coimmunoprecipitation. The presence of an equimolar concentration of HK did not affect FXII binding to uAPR in the presence of zinc, while the excess molar concentration of HK reduced the binding even in the presence of zinc (Supplementary Fig. S11h). Taken together, these data support a model in which the local expression of FXII in renal tubular cells in early DKD initiates uPAR signaling and thus promotes DNA damage.

## FXII interacts with multiple sites on uPAR through its heavy chain domains

To identify the interacting residues of FXII and uPAR, we performed computational modeling of the heavy chain of FXII and uPAR using AlphaFold2_multimer_v3 (see Supplementary Methods), which identified candidate binding residues in the fibronectin type II (FN2) and Kringle domains of FXII and corresponding residues in domains 1 and 2 of uPAR (Fig. 7a and Supplementary Table S4). To confirm the importance of the identified FXII residues, we designed sequential peptides covering these residues on the FN2 (HR13: [54]HRQLYHKCTHKGR[66]) and Kringle (TY10: [246]TYRNVTAEQA[255], PW15: [275]PWCFVLNRDRLSWEY[289]) domains (Supplementary Table S5). The area covered by the peptide HR13 derived from the FN2 domain of FXII has been previously shown to mediate binding to negatively charged surfaces[44]. Furthermore, HR13 shares sequence similarity in 8 amino acids out of 13 with the previously published peptide YHK9, which blocks FXII binding to HUVEC cells[2]. Pretreatment of HKC-8 cells with the peptides HR13 and PW15 (300 μM) reduced FXII binding to uPAR as determined by coimmunoprecipitation, while TY10 had no effect (Fig. 7b). Pretreatment with the peptides HR13 and PW15 reduced FXII-induced DNA damage and senescence suggesting a functional relevance of the amino acid stretches H54-R66 in FN2 domain and P275-Y289 in Kringle domain for FXII-induced uPAR signaling (Fig. 7c and Supplementary Fig. S12a). We next tested the stability of the newly generated peptide PW15 that

prevented FXII binding to uPAR in our experimental conditions. The peptide showed high stability for up to 24 h in HKC-8 cell culture medium (Supplementary Fig. S12b, c). Treatment of HKC-8 cells simultaneously with the peptides HR13 and PW15 induced markers of DNA damage and senescence, while single peptides (HR13 or PW15) failed to induce this response (Supplementary Fig. S12d, e). These results suggest that the simultaneous interaction of FXII with different uPAR binding sites is required for signal transduction and induction of DNA damage and senescence and that a combination of FXII-derived peptides can mimic the effect. Furthermore, we have synthesized 3 uPAR-based peptides based on the identified amino acids required for the uPAR-FXII interaction, namely RL20 ([52]RLWEEGEELELVEKSCTHSE[71]) and DL19 ([96]DLCNQGNSGRAVTYSRSRY[114]) from domain 1 and DV20 ([146]DVVTHWIQEGEEGRPKDDRH[165]) from domain 2 (Supplementary Table S5). The area covered by the peptide DL19 has been reported previously to interfere with the binding of HK to uPAR[45]. In addition, the peptide DV20 shares some sequence similarities with the previously reported peptide IQE13, a peptide from uPAR domain 2 that binds integrins, in addition to sharing a few amino acid sequences with the previously reported peptides QCR20 and EEG20 from uPAR domain 2[3]. We used AlphaFold2_multimer_v3 to predict the binding of these newly synthesized peptides to the heavy chain of FXII. Global docking of the peptides using AlphaFold2_multimer_v3 predicted binding to the FXII interface for the 3 peptides (Supplementary Fig. S12f). Pretreatment of HKC-8 cells with the peptides DL19 and DV20 (300 μM) reduced FXII binding to uPAR as determined by coimmunoprecipitation, while RL20 had no effect in our cellular model (Fig. 7d). Pretreatment with the peptides DL19 and DV20 reduced the induction of DNA damage and senescence markers by FXII or by the combination of FXII-based peptides HR13 and PW15, suggesting functional relevance of the amino acid stretches D96-Y114 and D146-H165 in uPAR domains 1 and 2, respectively, for FXII-induced uPAR signaling (Fig. 7e and Supplementary Fig. S12g–i).

To further determine the relevance of the FN2 domain of FXII and domain 2 of uPAR for the observed effects, we first blocked domain 2 of the human uPAR using the previously reported peptide PGS20 (Supplementary Table-S5)[3], which reduced FXII-induced DNA damage and senescence in HKC-8 cells (Supplementary Fig. S13a, b). To confirm the role of the FXII's FN2 domain for binding to uPAR, we ablated FXII expression in HKC-8 cells (*F12*-null HKC-8) using CRISPR/Cas9 technology (Supplementary Fig. S13c) and transfected *F12*-null cells either with wild type FXII (WT-FXII) or a FXII deletion mutant lacking FN2 domain (ΔFib-II)[46] (Supplementary Fig. S13d). DNA damage and senescence markers were reduced in tubular cells transfected with ΔFib-II-FXII compared to WT-FXII transfected cells (Supplementary Fig. S13e, f). To determine whether the observed effects require FXII's proteolytic activity, we used two different approaches: i) by using the FXII cyclic peptide inhibitors FXII-618 and FXII-900[47,48] (Supplementary Table S5) or ii) by rescuing *F12*-null HKC-8 cells with the FXII Locarno mutant (lacking the enzymatic activity[49]). Markers of DNA damage and senescence were not affected by the cyclic inhibitors or the expression of the Locarno mutant (Supplementary Fig. S14). Furthermore,

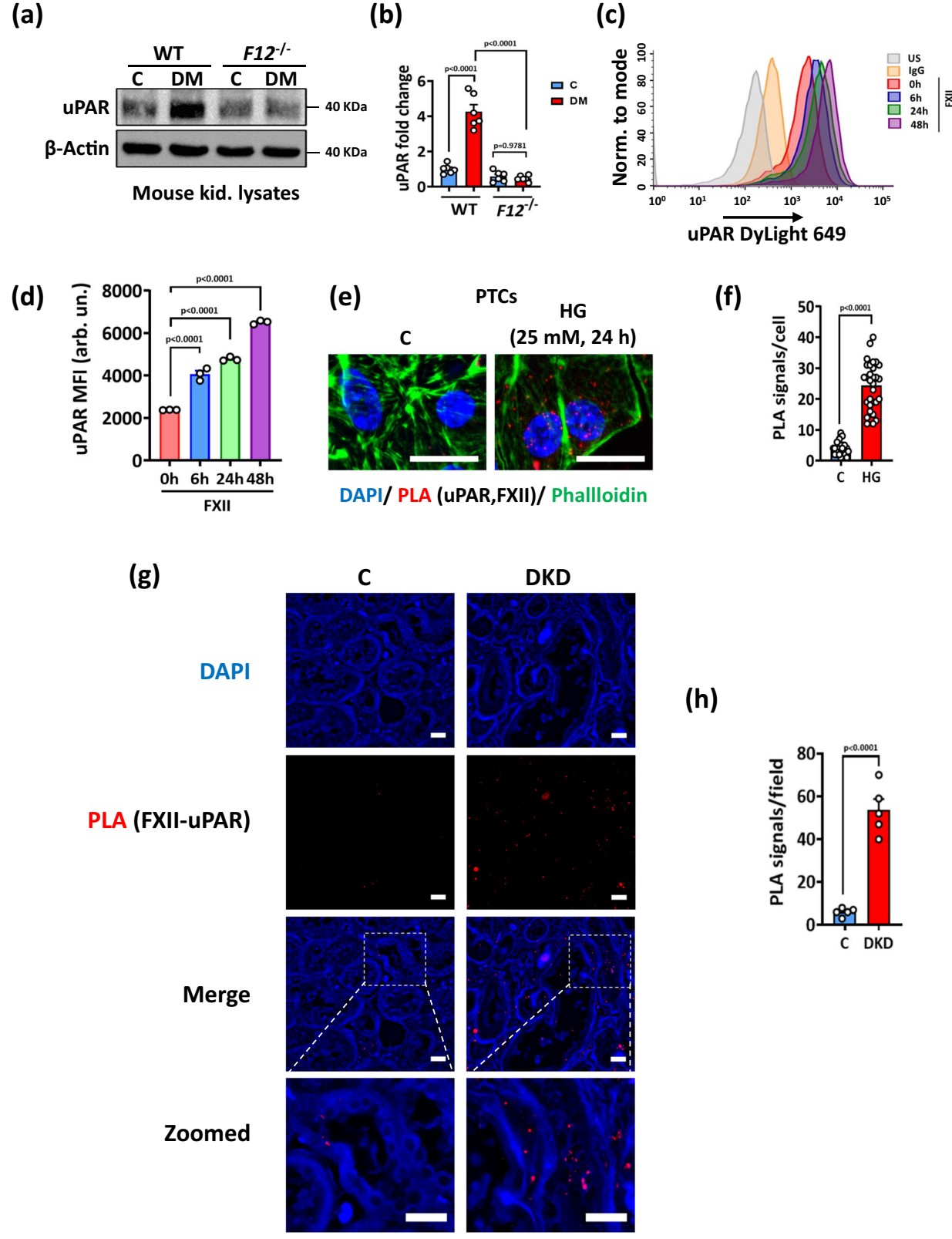

transcriptomic analysis of the Nephroseq® and the Karokidney public databases revealed that in addition to *KNG1* (Supplementary Fig. S11a, b), the expression of the contact pathway genes *KLKB1* (encoding kallikrein) and *F11* (encoding FXI) were downregulated in DKD patients (Supplementary Fig. S15a, b). We confirmed the downregulation of *Klkb1* and *Kng1* in hyperglycemic mice kidneys using qPCR and did not find a difference between WT and *F12⁻/⁻* mice (Supplementary Fig. S15c).

Furthermore, in hyperglycemic WT and *F12⁻/⁻* mice, mRNA levels of *F11* were comparable, as were D-dimer plasma levels, reflecting that coagulation activation was not different between genotypes (Supplementary Fig. S15c, d). Collectively, these results suggest a direct interaction between zymogen FXII and uPAR on kidney tubular cells, which regulates tubular uPAR expression and tubular cell senescence independent of FXII's proteolytic activity.

**Fig. 6 | FXII interacts with uPAR to signal on tubular cell surface.**
**a**, **b** Representative immunoblots (**a** loading control: β-Actin) and dot-plot summarizing results (**b**) for uPAR expression in kidney lysates of normoglycemic controls (C) and hyperglycemic (DM) wild type (WT) and *F12*⁻ᐟ⁻ mice. Dot-plot reflecting mean ± SEM of 6 mice per group; two-way ANOVA with Tukeys's multiple comparison test. **c**, **d** Representative histogram (**c**) and dot-plot summarizing the results (**d**) of uPAR surface staining determined by flow cytometry (mean fluorescence intensity, MFI) in HKC-8 cells exposed to purified human FXII (62 nM) in the presence of $Zn^{2+}$ (10 μM) for 6, 24, and 48 h. Dot-plot reflecting mean ± SEM of 3 independent experiments; one-way ANOVA with Tukeys's multiple comparison test. **e**, **f** Representative images of proximity ligation assay (PLA, **e**) and dot-plot summarizing results (**f**) in PTCs exposed to normal (5 mM) or high (25 mM) glucose

for 24 h. PLA signals representing FXII and uPAR interaction are immunofluorescently detected, red; DAPI nuclear counterstain, blue; phalloidin for cytoskeleton, green. Scale bars represent 20 μm. Dot-plot reflecting mean ± SEM of 3 independent experiments quantifying 30 cells from each condition with each dot representing the number of PLA signals/cell; two-tailed unpaired student's *t* test. **g**, **h** Representative histological images of proximity ligation assay (**g**, PLA, red dots representing FXII and uPAR interaction) and dot-plot summarizing results (**h**) in human kidney sections of nondiabetic controls (C) or diabetic patients with DKD (DKD); DAPI nuclear counterstain (blue). Scale bars represent 20 μm. Dot-plot reflecting mean ± SEM of 5 samples per group with each dot representing the mean of PLA signals/field for one sample; two-tailed unpaired student's *t* test. Source data are provided as a "Source Data" file.

## Integrin β1 is required for FXII signaling via uPAR in renal tubular cells

uPAR itself is not signaling competent and requires coreceptors such as integrins[50]. Functional annotations of the DEGs indicated FXII-dependent regulation of integrin signaling (Fig. 3d), and integrin β1 had the highest FDR among the downregulated integrins in hyperglycemic *F12*⁻ᐟ⁻ mice kidneys (Fig. 8a). The downregulation of several integrins in the kidneys of hyperglycemic *F12*⁻ᐟ⁻ mice was confirmed by qRT-PCR and immunoblotting (Fig. 8b and Supplementary Fig. S16a–c). To study a possible FXII-driven interaction of uPAR with integrins β1 and β3, two integrins known to interact with uPAR and are expressed by kidney cells[3,20,51], we exposed HKC-8 cells to FXII and performed coimmunoprecipitation assays. The uPAR-integrin β1 interaction increased upon exposure of HKC-8 cells to FXII, but binding to integrin β3 was not affected (Fig. 8c). The increased uPAR-integrin β1 interaction in FXII-treated HKC-8 cells was confirmed by PLA (Fig. 8d, e). To confirm whether FXII's FN2 domain is required to induce uPAR interaction with integrin β1, we transfected *F12*-null HKC-8 cells with WT-FXII or the ΔFib-II-FXII mutant. The latter reduced the interaction of uPAR with integrin β1 (Fig. 8f, g). Blocking integrin β1 with a monoclonal antibody abolished FXII-induced induction of γ-H2AX, p21, and KIM-1 in HKC-8 cells while blocking integrin β3 had no effect (Supplementary Fig. S16d–g). To determine whether uPAR-integrin β1 interaction is sustained in the tubular compartment under hyperglycemic conditions in vivo, we performed PLA on normoglycemic and hyperglycemic WT and *F12*⁻ᐟ⁻ mice. The uPAR-integrin β1 interaction was readily detectable in hyperglycemic WT but almost absent in hyperglycemic *F12*⁻ᐟ⁻ mice (Supplementary Fig. S17a, b). To validate these findings in human DKD, we conducted uPAR-integrin β1 PLA on kidney biopsies of nondiabetic controls and DKD patients. The strong interaction of uPAR and integrin β1 was readily detectable in the kidneys of DKD patients, which was accompanied by FXII upregulation (Fig. 8h and Supplementary Fig. S17c). Furthermore, immunostaining for FXII, uPAR, and integrin β1 showed colocalization of the three proteins in DKD patient biopsies compared to controls (Supplementary Fig. S18).

To identify the relevant integrin α-subunit interacting with integrin β1 upon FXII-uPAR stimulation in DKD, we focused on integrins α6 and α5, which were the most downregulated alpha subunits (highest FDR values next to integrin β1) in hyperglycemic *F12*⁻ᐟ⁻ mice kidneys compared to WT mice (Fig. 8a). To determine a possible role of integrins α6 and α5, we exposed HKC-8 cells to FXII and performed coimmunoprecipitation. While exposure of HKC-8 cells to FXII increased the integrin β1/α6 interaction, the integrin β1/α5 interaction was not affected (Supplementary Fig. S19a). To scrutinize whether the integrin α6β1 heterodimer mediates FXII-uPAR-dependent tubular injury and senescence, we blocked integrin α6 with a functional blocking monoclonal antibody (1 μg/ml). Blocking integrin α6 abolished FXII-induced induction of injury markers in HKC-8 cells (Supplementary Fig. S19b, c).

Aberrant integrin β1 signaling is associated with abnormal focal adhesions contributing to senescence[52]. To investigate whether signaling of FXII-uPAR-integrin β1 axis modulates focal adhesions, we

determined phosphorylation of focal adhesion kinase (FAK) and Src kinase in HKC-8 cells exposed to FXII. FXII time-dependently induced phosphorylation and activation of FAK and Src, which was associated with upregulation of the Rho family GTPase Rac1 and of the ROS regulator NADPH oxidase 1 (NOX1) (Supplementary Fig. S20a, b). To determine whether this pathway is activated upon FXII-uPAR interaction we pretreated cells with the inhibitory uPAR-based peptides DL19 and DV20 (300 μM) or a functional blocking monoclonal antibody targeting integrin β1 (10 μg/ml). Both interventions abolished FXII's effects on focal adhesion kinase activation or the upregulation of Rac1 and NOX1 (Supplementary Fig. S20c–f). Collectively, these data support a model in which the integrin α6β1 heterodimer mediates FXII-uPAR intracellular signaling via FAK-Src, thereby promoting DNA damage and senescence in kidney tubular cells.

## Targeting FXII as a therapeutic strategy for DKD

To determine whether reducing FXII levels could ameliorate kidney tubular senescence and experimental DKD, we treated mice after 16 weeks of persistent hyperglycemia with a *vivo* morpholino targeting FXII for an additional 6 weeks (Fig. 9a). The FXII *vivo* morpholino markedly reduced FXII expression in hyperglycemic kidneys compared to untreated mice or to mice treated with a scrambled mismatch morpholino (Supplementary Fig. S21a, b). The FXII *vivo* morpholino reduced kidney dysfunction, structural changes, DNA damage, and senescence in experimental DKD (Fig. 9b–e and Supplementary Fig. S21c–h). Importantly, targeting FXII expression reduced the uPAR-integrin β1 interaction in hyperglycemic mice (Supplementary Fig. S22). Thus, targeting FXII or the FXII-uPAR interaction may represent a therapeutic approach to reduce senescence and the associated progression of DKD.

## Discussion

This study identifies a previously undescribed role of zymogen FXII signaling in the progression of DKD. Based on our results, we propose that the zymogen FXII binds to uPAR and signals via integrin β1 on tubular cells, promoting DNA damage and senescence. The current results suggest that therapies targeting FXII or the FXII-uPAR interaction constitute modern strategies to ameliorate DKD. Further preclinical and translational studies are required to validate the therapeutic potential of this identified role of FXII-uPAR signaling in DKD.

The current study identifies tubular uPAR as a receptor for the detrimental effects of FXII. uPAR induction and signaling have been previously reported in human and murine DKD in glomerular cells[19,53,54]. We observed a strong induction of uPAR in tubular cells upon exposure to a hyperglycemic milieu, which was not apparent in hyperglycemic *F12*⁻ᐟ⁻ kidneys. Moderately increased uPAR expression upon exposure to FXII has been previously shown in neutrophils[55]. The mechanism through which FXII regulates uPAR surface expression remains to be fully understood.

uPAR is expressed by senescent cells and has been considered as a therapeutic target for senolytic strategies[18]. Here we propose a mechanism through which uPAR induces senescence, thus identifying

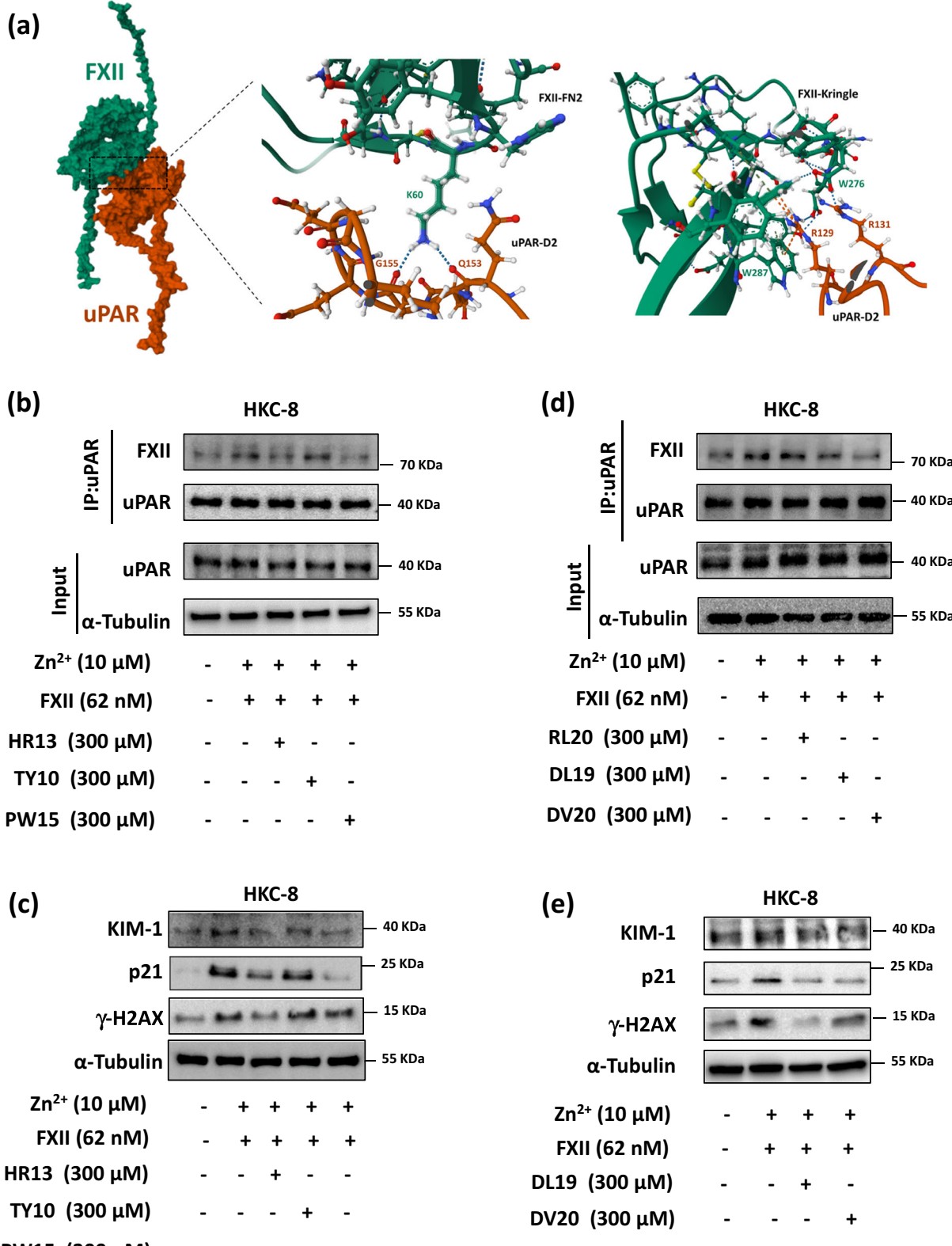

previously undescribed approaches to therapeutically target uPAR-mediated senescence. FXII-induced uPAR expression on tubular cells was associated with ROS accumulation, DNA damage, and senescence in human and murine kidney tubular cells. Kidney tubular cells that are normally resting in the G1 phase of the cell cycle respond to DNA-damaging stressors under hyperglycemic conditions, aiming to maintain genomic stability, by inducing cell cycle inhibitors and cell cycle

checkpoint regulators[56,57]. However, persistent hyperglycemic stress beyond the cellular repair capacity drives premature senescence[58]. Expression profiling revealed that the protection against kidney injury in hyperglycemic *F12*[-/-] mice was linked to negative enrichment of key pathological pathways related to DKD, including cell cycle arrest. Based on the mouse model used and the clinical data, we speculate that FXII-uPAR signaling is an early event in the course of DKD leading

**Fig. 7 | FXII interacts with uPAR through its FN2 and kringle domains.**
**a** Computational model of the FXII-uPAR complex with magnified areas depicting the interaction of FXII's fibronectin type II (FN2) and Kringle domains with uPAR domain 2 at multiple sites. **b** Representative immunoblots for FXII and uPAR from uPAR coimmunoprecipitation (IP, top) and immunoblots for uPAR, and α-Tubulin from the input (input, bottom) of HKC-8 cells exposed to purified human FXII (62 nM) in the presence of $Zn^{2+}$ (10 μM) for 24 h with and without pretreatment with FXII sequential peptides (HR13, TY10 and PW15; 300 μM) for 1 h compared to control non-treated cells (C). Input serves as a loading control. Immunoblots represent 3 independent experiments. **c** Representative immunoblots (loading control: α-Tubulin) for γ-H2AX,

p21, and KIM-1 expression in experimental groups (as described in **b**). Immunoblots represent 3 independent experiments. **d** Representative immunoblots for FXII and uPAR from uPAR coimmunoprecipitation (IP, top) and immunoblots for uPAR, and α-Tubulin from the input (input, bottom) of HKC-8 cells exposed to purified human FXII (62 nM) in the presence of $Zn^{2+}$ (10 μM) for 24 h with and without pretreatment with uPAR based peptides (RL20, DL19, and DV20; 300 μM) for 1 h compared to control non-treated cells. Input serves as a loading control. Immunoblots represent 3 independent experiments. **e** Representative immunoblots (loading control: α-Tubulin) for γ-H2AX, p21, and KIM-1 expression in experimental groups (as described in **d**). Immunoblots represent 3 independent experiments.

to tubular ROS and senescence. We cannot exclude that at advanced stages, excess molar concentrations of competing proteins, such as HK, may cross the dysfunctional glomerular filtration barrier and modulate FXII-uPAR signaling. Whether this would have an impact on already established tubular senescence remains to be studied.

Senescent cells are characterized by their adhesive phenotype, which is associated with increased focal adhesions and reduced motility[59,60]. Furthermore, secreted metalloproteinases in the SASP modulate the extracellular matrix (ECM) of senescent cells and cause tissue remodeling[61]. Consistently, pathways related to ECM remodeling and integrin signaling, which are typically induced in DKD[62], were negatively enriched in hyperglycemic *F12*[-/-] mice. Reducing FXII levels in vivo attenuated premature senescence and the associated SASP. Taken together, our data suggest that locally produced FXII interacts with uPAR and induces tubular senescence and associated changes, leading to kidney damage.

Signaling effects of the zymogen FXII have been linked to its interaction with uPAR via its heavy chain, which mediates binding and uPAR-dependent signaling[3,4,63]. The heavy chain domains protect the circulating zymogen FXII from autoactivation through steric hindrance provided by the interaction between the FN2 and the Kringle domains, keeping FXII in a closed conformation[46,64–66]. In response to surface binding, this closed conformation is relaxed, exposing the cleavage site for activation[64,65]. Considering the importance of FXII heavy chain domains for surface interactions, we focused on the heavy chain for computational structural modeling and identified candidate binding residues in FXII's FN2 and Kringle domains and corresponding residues on domains 1 and 2 of uPAR mediating the FXII-uPAR interaction. This modeling and our experimental data identify new molecular interactions of FXII and uPAR, extending previous findings[3]. Furthermore, the blocking effects of the peptides DL19 and PW15 derived from uPAR domain 1 and FXII's Kringle domains, respectively, suggest previously undescribed binding sites of the FXII-uPAR interaction. Previous studies suggested the involvement of FXII's Kringle domain in binding to artificial surfaces[67]. Binding of the Kringle domain of FXII to domains 1 and 2 of uPAR seems possible, taken into account the following evidence: (i) the important role of the Kringle domain of urokinase-type plasminogen activator (uPA) to uPAR binding[68], (ii) the high sequence similarity between the Kringle domains of FXII and uPA, and (iii) the inhibition of this binding by peptides targeting uPAR domains 1 and 2[69]. Our computational docking predicted the binding of the synthesized peptides to their corresponding residues, yet not all of them showed inhibitory effects on the FXII-uPAR interaction in our experiments. These differences may reflect discrepancies between the in silico and in vitro approach or indicate that the interacting residues between FXII and uPAR depend on the specific cell type and experimental conditions. Interestingly, the experimental results indicate that the simultaneous interaction of FXII with different uPAR binding sites is required for signal transduction and induction of DNA damage and senescence. Accordingly, blocking one binding site is sufficient to inhibit FXII's effect. Further detailed analyses of the FXII-uPAR interaction may hence identify new molecular targets, allowing to therapeutically modulate this interaction.

In addition to reduced tubular injury, glomerular injury was ameliorated in hyperglycemic *F12*[-/-] mice in the current study. Glomerular injury is likewise reduced in *F12*[-/-] mice with sickle cell disease[55], but the authors did not investigate tubular injury. The direct effect of zymogen FXII on podocytes or other glomerular cells seems possible, given the established role of uPAR in glomerular pathologies[19,20]. Yet, the relation of tubular and glomerular injury is bidirectional and some studies suggest that tubular damage is in part independent of and may even precede glomerular injury in DKD[70–72]. We observed FXII expression in kidney tubular cells, which increased in response to high glucose in vitro and in vivo. Local FXII production by cells other than hepatocytes[4,6,73] (and human protein atlas) and FXII-mediated effects independent of hepatic FXII have been reported[4], suggesting that tubular-released FXII may induce tubular damage in an auto- or paracrine fashion. Based on the current results we assume that the protective phenotype observed with FXII morpholino treatment as reflected by reduced tubular senescence and ameliorated albuminuria reflects the contribution of tubular cell injury to albuminuria in DKD[74]. The precise role of tubular FXII in kidney injury, e.g., a differential effect in acute versus chronic renal injury, requires further investigation, including, for example, mouse models with cell-specific FXII-inactivation.

Increased tubular FXII expression is expected to increase urinary FXII levels. We show that urinary FXII levels correlated with increasing severity of DKD in two independent cohorts of type-2 diabetic patients, indicating the utility of urinary FXII as a tubular injury biomarker. We currently do not know whether the increased urinary FXII selectively results from increased tubular FXII expression or increased filtration of plasma-derived FXII. Given its size (80 kDa), FXII may cross the glomerular filtration barrier once barrier dysfunction is established. In addition, while our data propose that urinary FXII reflects tubular injury, the levels measured in urine most likely do not reflect levels reached locally in the tubular compartment or at the surface of tubular cells. Further studies are required to elucidate the exact role and the regulation of tubular FXII in the pathogenesis of DKD. Another interesting question is whether the recently identified nephroprotective therapies (SGLT2-inhibitors and GLP-1 agonists) regulate tubular FXII expression. These questions need to be addressed in the future.

Consistent with previous reports showing that zinc is a cofactor required for FXII surface binding[2,4,55,75], the FXII-uPAR dependent effects on tubular cells were found to be zinc-dependent. In blood, activated platelets release zinc from internal stores[76]. Furthermore, activated platelets are known to contribute to DKD[77], but it remains currently unclear how zinc released from the platelets may reach the intratubular lumen to facilitate FXII-uPAR binding. Alternatively, renal tubular cells express zinc transporters and reabsorb zinc[78]. In addition, deficiency of metallothioneins, a family of heavy metal binding proteins expressed by tubular cells and maintaining zinc homeostasis, exacerbates DKD in murine models by inducing oxidative stress[79,80]. Therefore, it appears possible that tubular injury in hyperglycemic conditions releases zinc from its intracellular stores, enhancing FXII binding to uPAR and, hence FXII signaling. Indeed, urinary excretion of zinc is increased in CKD patients[81], which may provide sufficient zinc in the tubular lumen. The origin of zinc needs to be characterized in the

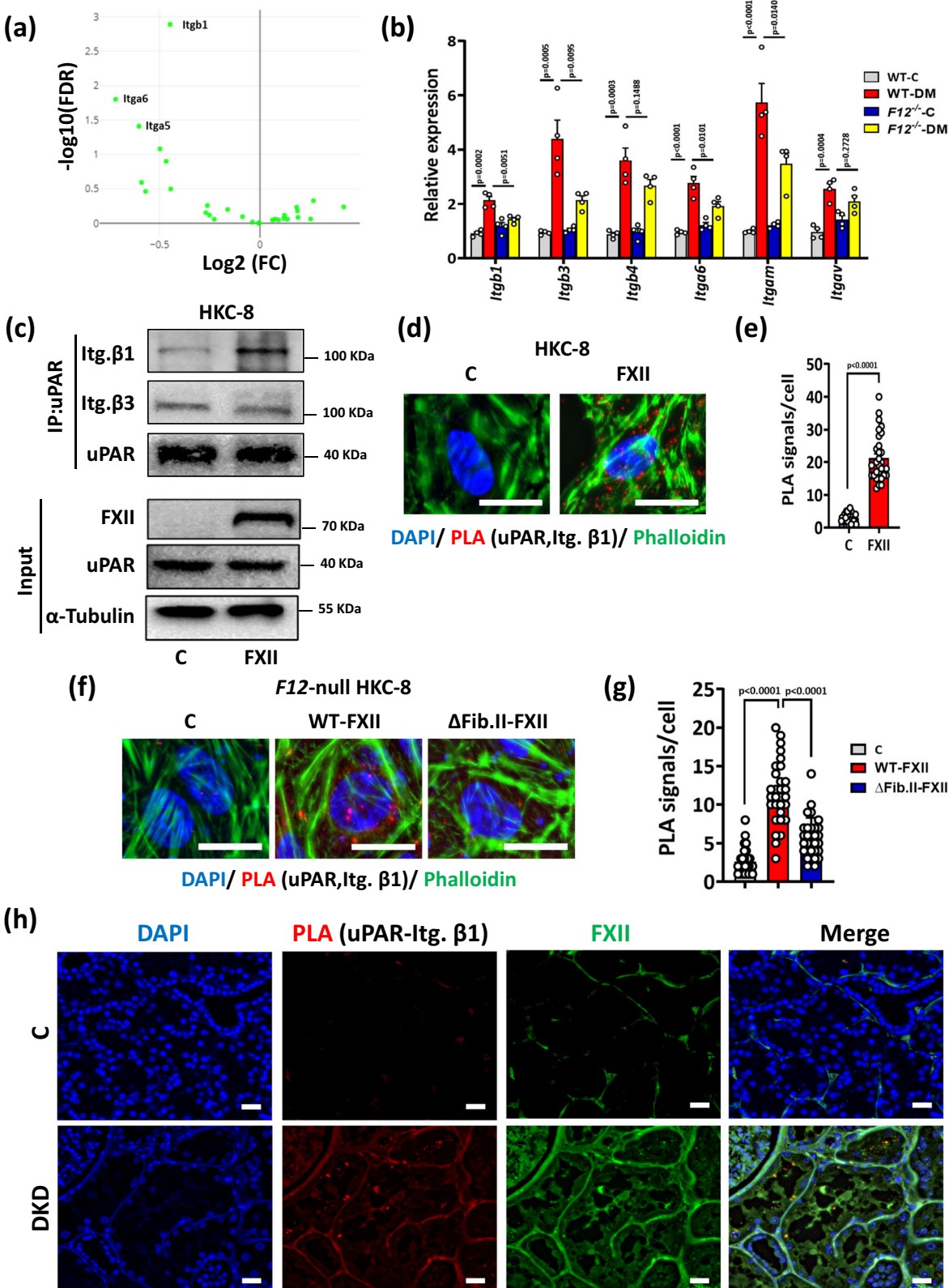

future and may provide new therapeutic options to restrict the FXII-uPAR interaction on tubular cells.

Zinc mediates FXII surface binding but may also facilitate FXII activation. Our data suggest that the effects of FXII on tubular cells are independent of its activation, in line with previous reports on other cell types[2-4]. Furthermore, the differential regulation of the contact pathway proteins in the kidneys of DKD patients and in hyperglycemic mice

and the absence of coagulation changes in $F12^{/-}$ mice compared to WT mice support the notion that the effect of FXII is independent of coagulation activation. Data describing a function of the KKS, which is activated by the protease FXII, in kidney injury are controversial as earlier studies suggested that bradykinin promotes tubular and glomerular injury[82,83], while other more recent studies imply protective effects of exogenous kallikrein and bradykinin receptors in DKD[84,85].

**Fig. 8 | Integrin β1 is required for FXII-uPAR signaling on tubular cells.**
**a** Volcano plot comparing hyperglycemic *F12⁻/⁻* mice to hyperglycemic WT mice based on Log fold change (FC) values and the false discovery rate (FDR); integrins are shown in green. Integrin β1 (*Itgb1*) is the most downregulated integrin in hyperglycemic *F12⁻/⁻* mice kidneys by FDR. **b** Bar graphs summarizing the expression (qRT-PCR) of selected integrin genes comparing normoglycemic controls and hyperglycemic WT and *F12⁻/⁻* mice. Bar graphs reflecting mean ± SEM of 4 mice per group; two-way ANOVA with Tukeys's multiple comparison test. **c** Representative immunoblots for active integrin β1, active integrin β3, and uPAR from uPAR coimmunoprecipitation (IP, top) and immunoblots for FXII, uPAR, and α-Tubulin from the input (input, bottom,) of HKC-8 cells exposed to purified human FXII (62 nM) in the presence of Zn²⁺ (10 μM) for 24 h (FXII) compared to control non-treated cells (C). Input serves as a loading control. Immunoblots represent 3 independent experiments. **d, e** Representative images of proximity ligation assay (PLA, **d**; red, uPAR and active integrin β1 interaction) and dot-plot summarizing results (**e**) in experimental groups (as described in **c**); DAPI nuclear counterstain, blue; phalloidin for cytoskeleton, green. Scale bars represent 20 μm. Dot-plot reflecting mean ± SEM of 3 independent experiments quantifying 30 cells from each condition with each dot representing the number of PLA signals/cell; two-tailed unpaired student's *t* test. **f, g** Representative images of proximity ligation assay (PLA, **f**; red dots representing uPAR and active integrin β1 interaction) and dot-plot summarizing results (**g**) in *F12*-null HKC-8 cells transfected with wild type FXII (WT-FXII) or a FXII deletion mutant lacking fibronectin type II domain (ΔFib-II-FXII) compared to empty-vector transfected cells (C, controls); DAPI nuclear counterstain, blue; and phalloidin for cytoskeleton, green. Scale bars represent 20 μm. Dot-plot reflecting mean ± SEM of 3 independent experiments quantifying 30 cells from each condition with each dot representing the number of PLA signals/cell; one-way ANOVA with Tukeys's multiple comparison test. **h** Representative images of proximity ligation assay (PLA, f; red dots representing uPAR and active integrin β1 interaction) in human kidney sections of nondiabetic controls (C) or diabetic patients with DKD (DKD); DAPI nuclear counterstain, blue; and FXII, green. Scale bars represent 20 μm. Source data are provided as a "Source Data" file.

---

Our data suggest a detrimental effect of zymogen FXII which is hence independent of the KKS. Further analyses are required to delineate a possible additional effect of the activated FXII in DKD.

The lack of a transmembrane domain in uPAR requires its association with membrane spanning receptors, such as integrins[50,86]. Integrins form complexes with FXII and uPAR, promoting FXII-dependent signaling in a cell and context specific manner[87,88]. Upregulation of integrins such as β1, β3, and β6 is linked to EMT, fibrosis, and senescence in injured tubular cells[51,89,90]. Persistent integrin β1 signaling is associated with sustained focal adhesions and ROS production that contribute to senescence[52]. In epithelial cells, the most abundant integrins are β1-containing heterodimers, and integrin β1 signaling can induce diverse cellular responses depending on the cell type, the α subunit forming the integrin heterodimer, the binding ligand, and the cellular microenvironment[91–93]. Our results showed (i) that the uPAR/FXII complex required active integrin β1 but not β3, that (ii) a blocking antibody against integrin β1 almost completely prevented FXII-induced tubular cell injury, that (iii) the integrin β1 forms a heterodimer with integrin α6, which promotes FXII-associated DNA damage and tubular cell injury. The integrin α6β1 heterodimer induces ROS, DNA damage and senescence in human fibroblasts[94], similar to the phenotype we observed in kidney tubular cells in our study. Integrin signaling activates the Rho family GTPase Rac1 which is promoted by FAK[94–96]. Rac1 activates NADPH oxidases (NOXs), increasing ROS generation, DNA damage and senescence[97,98]. Constitutive integrin signaling associated with abnormal focal adhesions induces senescence through increased ROS production[52]. Integrin β1 signaling mediates cellular adhesion through phosphorylation of FAK, and the latter acts as a scaffolding platform for other kinases including Src kinase[95,99]. Src activation maintains a senescence phenotype in fibroblasts in response to DNA damage[100]. In the current study, FXII deficiency was associated with the downregulation of pathways related to integrin signaling, focal adhesions, and Rho GTPase signaling in hyperglycemic kidneys. Furthermore, interference with FXII-uPAR binding or blocking integrin β1 reduced the activation of focal adhesion kinases and the upregulation of Rac1 and NOX1 in tubular cells, suggesting the involvement of abnormal focal adhesions in FXII-mediated oxidative DNA damage. Collectively, these results are consistent with a model in which FXII-uPAR interaction induces tubular senescence via integrin β1 signaling. Considering the wide range of functions and effects of integrins, therapies targeting upstream receptor mechanisms, such as the FXII-uPAR interaction, could be superior approaches to target integrin-mediated detrimental effects.

In conclusion, our findings show a previously undescribed function of FXII in the progression of DKD in which FXII-dependent stabilization of and interaction with uPAR on tubular cells induces integrin β1 signaling that in turn promotes DNA damage and senescence.

Targeting FXII or its interaction with uPAR may represent promising therapeutic avenues for DKD.

## Methods

The research conducted in this study complies with the ethical regulations of the University of Leipzig (Ethic vote no: 263-2009-14122009 and 201/17-ek), the University of Heidelberg (Ethic vote no: S-383/2016), the confirmation of the related notification by the local animal care and use committee (T01_20; Landesdirektion, Leipzig, Germany), the local animal care and use committee (Landesverwaltungsamt, Halle, Germany), and the local animal care and use committee (Heidelberg, Germany).

### Study design

The objectives of this study were to identify the possible role of FXII in the pathogenesis of DKD and to elucidate the involved mechanism. FXII expression was analyzed in human kidney biopsies (immunostaining), in urine samples (ELISA) and in various murine models of DKD (streptozotocin induced DKD, db/db mice). Kidneys of WT and *F12⁻/⁻* mice were analyzed using RNA-sequencing, immunostaining, and immunoblotting. Furthermore, time kinetics and mechanistic studies were conducted in human and murine kidney tubular cell lines in vitro. In silico computational structural modeling of the interacting proteins was conducted to elucidate the signaling effect of FXII. Details of all experimental procedures are provided in the supplementary methods.

### Human kidney biopsies and urine samples

Human kidney biopsies of type-2 diabetic patients with established DKD and non-diabetic controls were obtained from the tissue bank of the National Center for Tumor Diseases, Heidelberg, Germany (Supplementary Table S1). Human urine samples of control and type-2 diabetic individuals with different CKD stages were obtained from the LIFE-ADULT cohort (Ethic vote no: 263−2009−14122009 and 201/17-ek, University of Leipzig, Supplementary Table S2)[101] and from the Heidelberger Study on Diabetes and Complications (HEIST-DiC, Ethic vote no: S-383/2016, University of Heidelberg, Supplementary Table S3)[102]. In both cohorts, CKD severity was classified according to the KDIGO criteria[103].

### Murine model of DKD and in vivo interventions

Male wild-type (WT) C57BL/6 mice, nondiabetic C57BLKsJ-db/+ (db/m), and diabetic C57BL/KsJ-db/db (db/db) mice were obtained from Janvier (S.A.S., St. Berthevin Cedex, France). Male *F12* deficient (*F12⁻/⁻*) mice on C57BL/6 background were provided by Thomas Renné and have been previously described[104]. Mice were maintained at a temperature of 21 ± 2 °C and a humidity of 55 ± 15% with free access to standard chow diet (V1534-0, ssniff, Germany) and water. Persistent hyperglycemia was induced in male WT and *F12⁻/⁻* mice at the age of 8 weeks using intraperitoneal injections of streptozotocin (STZ) freshly dissolved in 0.05 mM sterile sodium citrate (pH 4.5, 60 mg/kg body weight for 5

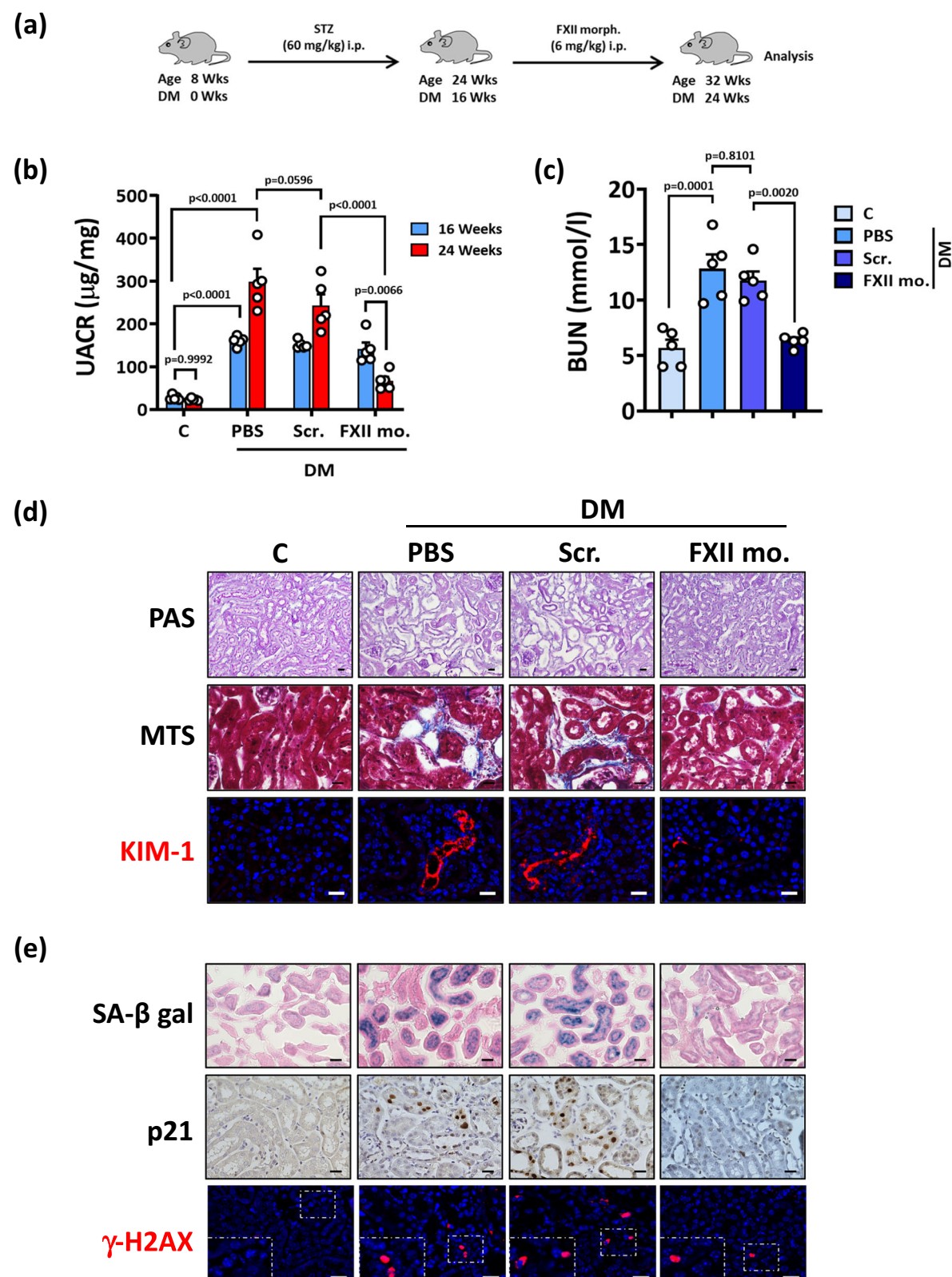

consecutive days)[105]. Age-matched control mice of both genotypes received intraperitoneal sodium citrate injections. Blood glucose levels were determined routinely using ACCU-CHEK glucometer with blood samples collected from the tail vein. Mice that had a blood glucose level above 17 mM after 2 weeks of STZ injections were considered diabetic. Mice that had a blood glucose level of more than 28 mM received 1–2 units of insulin glargine (Lantus®; SANOFI, France) subcutaneously to avoid potentially lethal hyperglycemia. Mice were maintained under persistent hyperglycemia with continuous monitoring for 24 weeks, then they were sacrificed to collect blood and organs for further analyses.

A subset of WT mice were treated after 16 weeks of persistent hyperglycemia with a *vivo* morpholino oligomer (5'-ACCCCAGGAA CAACAGAGCCGTCAT-3') targeting the translation of mouse *F12* gene (6 mg/kg body weight in PBS, every other day, intraperitoneal) or a

**Fig. 9 | Targeting FXII ameliorates established DKD. a** Experimental scheme of the DKD model with interventions. WT mice were age-matched, and persistent hyperglycemia was induced using streptozotocin (STZ) and maintained for 16 weeks. Subgroups of mice were treated with PBS, mismatch morpholino, or FXII translational blocking morpholino (FXII morph.) starting from 16 weeks of hyperglycemia for a further 6 weeks. Wks: weeks. **b** Dot-plot showing average urinary albumin-creatinine ratio (UACR, µg albumin/mg creatinine) in experimental groups (as described in **a**) after 16 or 24 weeks of persistent hyperglycemia. Dot-plot reflecting mean ± SEM of 5 mice per group; two-way ANOVA with Tukeys's multiple comparison test. **c** Dot-plot summarizing blood urea nitrogen (BUN, mmol/l) in experimental groups (as described in **a**). Dot-plot reflecting mean ± SEM of 5 mice per group; one-way ANOVA with Tukeys's multiple comparison test. **d** Exemplary

histological images of periodic acid Schiff staining (top panel, PAS), interstitial fibrosis (middle panel, Masson's trichrome stain, MTS), and kidney injury molecule-1 immunostaining (bottom panel, KIM-1, red; DAPI nuclear counterstain, blue) in experimental groups (as described in **a**); scale bars represent 20 µm. **e** Exemplary histological images of senescence-associated β-galactosidase (top panel, SA-β-gal, blue; eosin counterstain), p21 immunostaining (middle panel, detected by HRP-DAB reaction, brown; hematoxylin nuclear counter stain, blue), and phosphorylated H2A histone X (bottom panel, γ-H2AX, immunofluorescently detected, red; DAPI nuclear counterstain, blue, insets show higher magnification of the marked areas) in experimental groups (as described in **a**); scale bars represent 20 µm. Source data are provided as a "Source Data" file.

mismatch *vivo* morpholino (5'-CCCCGCTGCCTGCCCAGGA-3') for further 6 weeks according to an established protocol[106]. Control mice received PBS every other day (intraperitoneal) for the same duration. All animal experiments were conducted following standards and procedures approved by the local animal care and use committee (animal proposal number: TVV 70/21, Landesverwaltungsamt, Leipzig, Germany).

### Cell culture and in vitro interventions

Immortalized human-derived renal proximal tubular cells (HKC-8) were cultured in a mixture of DMEM glucose-free medium and Ham's F12 nutrient mixture in a ratio of 1:1 to achieve a final glucose concentration of 5 mM in the medium. The medium mixture was supplemented with 10% FBS, and the cells were maintained at 37 °C. Mouse primary proximal tubular cells (PTCs) were isolated and cultured according to an established protocol[107]. Freshly isolated mouse kidneys were decapsulated and the cortical area was separated and minced into small pieces and collected in HBSS containing Collagenase-I at a concentration of 200 units/ml. The small cortical fragments were digested at 37 °C for 30 min and the proximal tubule fragments were enriched using horse serum. The Proximal tubule containing fragments were pelleted down and washed twice with HBSS, then the pellet was resuspended in DMEM:F12 medium in a ratio of 1:1, supplemented with insulin/transferrin/selenium (Invitrogen, 5 µg/ml, 2.75 µg/ml and 3.35 ng/ml respectively), APO transferrin (Sigma-Aldrich, 2.0 µg/ml), hydrocortisone (Sigma-Aldrich, 40 ng/ml), recombinant human epidermal growth factor (rhEGF, R&D Systems, 0.01 µg/ml), and 1% antibiotic/antimycotic solution (Sigma-Aldrich, 10,000 units/ml penicillin, 0.1 mg/ml streptomycin and 0.25 µg/ml amphotericin B). The medium was changed after 72 h, and the cells were maintained in the same culture medium without rhEGF.

For dose selection of FXII, HKC-8 cells were exposed to increasing concentration of purified human FXII (30, 62, or 150 nM) in the presence of 10 µM of $Zn^{2+}$ for 24 h. For the time kinetics study, HKC-8 cells or PTCs were exposed to purified human FXII (62 nM) or recombinant mouse FXII (62 nM), respectively, in the presence of 10 µM of $Zn^{2+}$ for 6, 24, and 48 h. In a subset of experiments, HKC-8 cells were pretreated with the human uPAR domain-1 based peptides RL20 and DL19, domain-2 based peptides DV20 and PGS20[3] or the human FXII-based peptides (HR13, TY10, and PW15), at a concentration of 300 µM for 1 h followed by treatment with purified human FXII (62 nM) in the presence of 10 µM of $Zn^{2+}$ for 24 h. In another set of experiments, HKC-8 cells were pretreated with the peptides HR13 and PW15 (300 µM) alone or in combination for 24 h with and without pretreatment with the peptides DL19 and DV20 for 1 h (300 µM). For FXII inhibitor experiments, HKC-8 cells were pretreated with the activated FXII cyclic peptide inhibitors, FXII-618[48] and FXII-900[47], at a concentration of 10 µM for 30 min, followed by treatment with purified human FXII (62 nM) in the presence of 10 µM of $Zn^{2+}$ for 24 h. For blocking integrins β1, β3, and α6 experiments, HKC-8 cells were preincubated with the function-blocking monoclonal antibodies against integrin β1 (clone P5D2)[108], integrin β3 (clone B3A)[109] at 10 µg/ml, or integrin α6

(clone GOH3)[110] at 1 µg/ml for 30 min followed by treatment with purified human FXII (62 nM) in the presence of 10 µM of $Zn^{2+}$ for 24 h.

### Generation of *F12*-null HKC-8 cells and FXII mutant transfection

HKC-8 cells were seeded in a 6-welll plate at a density of $4 \times 10^5$ cells/well. Confluent cells were transfected with GFP-expressing human FXII CRISPR/Cas9 KO plasmid (sc-409611; Santa Cruz Biotechnology) using Turbofect transfection reagent following manufacturer's protocol. After 24 h, transfected cells were sorted, and GFP-positive single cells were collected in 96-well plates. Single-cell clones were expanded and screened for FXII expression by immunoblotting. In different sets of experiments, *F12*-null HKC-8 cells were transfected with wild-type human FXII (pcDNA3-FXII WT), a FXII mutant lacking the FN2 domain (pcDNA3-FXII-ΔFibII), a FXII mutant lacking enzymatic activity (pcDNA3-FXII-Locarno), or empty vector (pcDNA3.1 +) using Turbofect transfection reagent following manufacturer's protocol. After 24 h, proteins were isolated for immunoblotting or cells grown on cover slips were fixed for staining.

### Statistical analysis

The data are summarized as mean ± SEM (standard error of the mean). Statistical analyses were performed with parametric Student's *t* test, one-way ANOVA, two-way ANOVA, or non-parametric Mann–Whitney and Kruskal–Wallis test, as appropriate, and post-hoc comparison with the method of Tukey or Dunnett's multiple comparisons. The Kolmogorov–Smirnov (KS) test or D'Agostino–Pearson-Normality-test was used to determine whether the data were consistent with a Gaussian distribution. Prism 9 (www.graphpad.com) software was used for statistical analyses. Statistical significance was accepted at values of $P < 0.05$.

### Reporting summary

Further information on research design is available in the Nature Portfolio Reporting Summary linked to this article.

## Data availability

The raw data from the RNA-seq analyses has been uploaded to the NCBI GEO database as BioProject under accession number PRJNA1064044. Patient data, the sequences of the used peptides and primers, in addition to the detailed experimental procedures are provided in the supplemental data file. Source data are provided in this paper.

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

## Acknowledgements

This work was supported by grants of the 'Deutsche Forschungsgemeinschaft' (DFG, German Research Foundation): IS-67/16-1, IS-67/22-1, IS-67/25-1, IS-67/26-1, SFB1423/C07 and EFSD (European Foundation for the Study of Diabetes) to B.I.; SH 849/1-2 and EFSD to K.S. (Khurrum Shahzad); KO 5736/1-1 and KO 5736/5-1 to S.K. (Shrey Kohli); and by DAAD scholarship to A.E. We thank Kathrin Deneser, Rumiya Makarova, Silke Borchert, Susann Kostmann, and Estela Mena Plaza for the excellent technical support.

## Author contributions

A.E. designed, performed, and interpreted in vivo, in vitro, and ex vivo experiments. D.G., R.R., J.M., S.A., S.F., S.Z., A.G., A.M., and Z.L. performed and assisted in vitro and ex vivo experiments. M.M.A.D. assisted in in vivo experiments. S.P. performed bioinformatic analyses. A.E., D.G., and K.S. (Kunal Singh) performed and assisted in FACS surface staining and cell cycle analysis. A.E., R.G., and S.K. (Shrey Kohli) conducted expression analysis and assisted in data interpretation. A.E., J.M., and R.R. conducted functional annotations and pathway enrichment analysis. A.S., S.K. (Stefan Kopf), and R.B. provided control and DKD urine samples. C.S. provided control and DKD kidney biopsies. T.G., B.A., and C.L. provided FXII cyclic peptide inhibitors, synthesized the peptides, and performed peptide stability assays. P.K. and G.K. conducted computational modeling studies and analysis. T.R. provided *F12-/-* mice and FXII mutant plasmids and reviewed the manuscript. K.S. (Khurrum Shahzad) assisted in interpreting the data and preparing the manuscript. A.E. and B.I. conceptually designed and interpreted the experimental work and prepared the manuscript.

## Funding

## Competing interests

The authors declare that they have no competing interests.
