## [Peer Review File · Nature Communications]

Factor XII signaling via uPAR-integrin β 1 axis promotes tubular senescence in diabetic kidney diseaseREVIEWER COMMENTS

Reviewer #1 (Remarks to the Author):

Overview.

This manuscript is a very interesting and moves the field forward.

Can the authors elaborate on why FXII appears to be a renal tubular toxin that stimulates DNA damage, oxidative effects, by the same pathway that it stimulates endothelial cell angiogenesis and neutrophil activity? Are FXII levels of 62 nM achievable in proximal tubule in diabetic nephropathy? Teleologically, why would PTCs have a pathway for injury if the agonist is not physiologically present? What is the physiologic role of FXII in the proximal tubule as you understand it from these elegant studies?

Although in fact I agree with the authors' finding that some FXII signaling pathway is mediated through uPAR and beta--1-integrin based on what is known with previously published work on endothelial cells, most investigators will not think it is convincing with just one inhibitory experiment with a peptide (PGS20) from domain 2 of uPAR. However, in glomerular-tubular fluid from a leaky glomerulus in a diabetic with Kimmelstiel-Wilson glomerulosclerosis, the plasma proteins high molecular weight kininogen, low molecular weight kininogen, vitronectin, and single chain urokinase may all be present. If so and in molar excess, can they block FXII-induced renal tubular cell toxicity inducing oxidative changes and DNA damage? Also, does FXII induce apoptosis in renal tubular cells? Apparently, apoptosis is not an entity in biologic senescence.

Last, I noticed that you used an older version of AlphaFold. Presently, AlphaFold2-multimer is an improvement on the original program. It would be of interest to see if the original data holds up and there are additional interactions observed.

Specific comments.

Introduction.

p.3, line 4 from the top. How does one define "senescence" in biologic terms? Is the reduction of leukocyte presence and activity a hallmark of biologic senescence?

p.3 line 12 from the top. You cite review articles by authors who mostly did not do the work. The citations that first outlined this pathway were Refs #35, 36, 18, and 60.

The Introduction overall rambles, i.e., does not show focus on the topic at hand. If this pathway was truly recognized by you in an unbiased way, your later review of the already published FXII literature on the function of zymogen FXII on endothelium, neutrophils, and lung fibroblasts helped confirm your assessment. Your introduction should communicate that knowledge already in the literature.

Methods.

p.11, lines 26, 27. What are the sequences for FXII peptides HR13, TY10, and PW15? No citation is provided in the manuscript text. The Supplementary table citation should be named in the main text. The original source of the peptides should be named, as well. Were they completed intuited from alpha fold, or did you see them in other literature sources as well? HR13 had previously been reported as an artificial surface binding site by Citarella et al. 2000 and shown to block FXII binding to HUVEC by Mahdi et al. 2002. Has PW15 been characterized before your alpha fold studies?

p.11, line 29. What is the citation or source of FXII cyclic peptide inhibitors; FXII-618, FXII-900? Again, the Supplementary table citation should be named in the METHODS. The original source should be named and put in the main references of the paper, as well.

Results.

p.5, lines 10,11. Collectively, it looks like the absence of FXII upregulates gene sets that could be related to diabetes and downregulates gene sets related to inflammation and cell function, i.e., slowing processes down. Your statement is not this clear.

p.5, lines 25,26. What is the source of the senescence gene set? Where is this group of genes or gene set derived?

Supplementary Figure S9d. Does peptide HR13, PW15, or combined HR13-PW15 alone stimulated gamma-H2AX, p21, or KIM-1 increase? This experiment is an appropriate control to the figure as presented. If so, is the pathway blocked uPAR peptide PGS20 and the peptide on beta-1-integrin (IQE13 - see LaRusch et. al.) that binds uPAR's domain 1?

p.7, lines 4-6. Were the authors able to confirm the downregulation of several integrins and the upregulation of beta-1-integrin from their RNA-Seq data, independent and in addition to their PCR data presented in Fig 7b?

Figure 7i. This very important figure would best be represented by a panel with DAPI alone, uPAR alone, ab Beta-1-integrin alone, and FXII alone and then uPAR-FXII, uPAR-beta-1-integrin, beta-1-integrin-FXII, and then DAPI/PLA (uPAR, beta-1-integrin/FXII). The whole interaction will be visualized.

p. 8, lines 42,43. Without question, your work has extended previous work, but as mentioned already above, peptide HR13 had previously been reported as an artificial surface binding site by Citarella et al. 2000 and shown to bind HUVEC by Mahdi et al. 2002.

p. 9, lines 24,25. Need to say that a local source, i.e., PTCs themselves when activated could be the source of the cofactor zinc ion.

p.9, line 29. Refs #35 and 36 before Ref 18 focused on examining the FXII zymogen and FXIIa issue in the FXII/FXIIa interaction with cells.

Fig 8b. This experiment shows that after diabetes, a morpholino for FXII knockdown in mice resulted in reduced albumin urinary loss at 24 weeks over 16 weeks. I understand how FXII could influence PTCs' activity via the uPAR-beta-1-signaling system, how does FXII control albumin levels in urine? Do proximal tubules influence albumin loss? If not, how do the authors explain this interesting result.

Reviewer #2 (Remarks to the Author):

The authors identify the role of zymogen FXII signaling in the progression of DKD suggesting that FXII binds to uPAR and signals via integrin β 1 on tubular cells, promoting DNA damage and senescence. They also indicate that therapies targeting FXII or the FXII-uPAR interaction constitute new strategies to ameliorate DKD. It is a well-designed and interesting study. I suggest acceptance of the paper after the following concerns being resolved.

1. The authors suggested that zymogen FXII induced DNA damage and senescence in tubular cells, whereas the involved mechanism is unknown.
2. The coagulation pathway and inflammatory effects of FXII which also contribute to progression of DKD might be checked.
3. the SASP should also be analyzed by ELISA besides q-pcr.

REVIEWER COMMENTS

Reviewer #1 (Remarks to the Author):

Overview.

This manuscript is a very interesting and moves the field forward.

We thank the reviewer for the thorough analysis of the manuscript and the supportive comments and we address the comments as follows:

Comment 1-1:

Can the authors elaborate on why FXII appears to be a renal tubular toxin that stimulates DNA damage, oxidative effects, by the same pathway that it stimulates endothelial cell angiogenesis and neutrophil activity?

Response 1-1:

We appreciate this interesting comment. We agree with the reviewer that the interaction of FXII with uPAR and signaling through integrins have been established in different cells promoting a variety of effects. Podocytes uPAR, which has been linked with DKD, interacts with integrin $\alpha\beta3$ causing podocyte effacement and albuminuria^{1,2}. Regulators of integrin $\beta1$ -dependent signaling include the type of α subunit forming the integrin heterodimer, the binding ligand, and the cellular microenvironment³⁻⁵. While previous reports and the human protein atlas indicate uPAR expression in kidney tubular cells in healthy and DKD patients^{6,7}, the role of tubular uPAR and its possible interactions with integrins in DKD has not been described hitherto. A role of integrin $\beta1$ in renal tubular cells has been established. Thus, integrin $\beta1$ expression is induced in proximal tubular cells (PTCs) after ATP depletion⁸. In ZSF1 rats (a rat model of DKD), integrin $\beta1$ expression is induced in PTCs and mesangial cells, and administration of a pan integrin antagonist ameliorated proteinuria and fibrosis⁹. These studies support a role of integrin $\beta1$ for kidney tubular cell injury, but the regulation and possible coreceptors of integrin $\beta1$ in this context remained unknown.

Based on these data we assume that context-specific effects of integrin $\beta1$ depend at least in part on the corresponding α subunit. To get further insights into the α subunit associated with integrin $\beta1$ in our study, we aimed to identify differentially expressed integrins based on the RNA-seq data. We identified integrin $\alpha6$ and $\alpha5$ as the most downregulated alpha subunits (highest FDR values next to integrin $\beta1$) in hyperglycemic *F12*^{-/-} mice kidneys compared to WT mice (**Fig. R1-1a**). To validate the relevance of these findings on protein level, we exposed HKC-8 cells to FXII and performed coimmunoprecipitation. The integrin $\beta1/\alpha6$ interaction increased upon exposure of HKC-8 cells to FXII, while the integrin $\beta1/\alpha5$ interaction was not affected (**Fig. R1-1b**). To determine whether the integrin $\alpha6\beta1$ heterodimer is involved in FXII-mediated tubular injury and senescence, we blocked integrin $\alpha6$ with a functional blocking monoclonal antibody (clone GOH3, 1 $\mu\text{g/ml}$). Blockage of integrin $\alpha6$ abolished FXII-mediated induction of $\gamma\text{-H2AX}$ and KIM-1 in HKC-8 cells (**Fig. R1-1c,d**).

Taken together, these data support a model in which FXII induces tubular injury and senescence through a signaling complex that involves uPAR and integrin $\alpha6\beta1$.

This finding is consistent with previous work linking integrin $\alpha6\beta1$ signaling to the induction of ROS, DNA damage, and senescence in human fibroblasts, similar to the phenotype we observed in kidney tubular

cells in our study ¹⁰. These findings indicate cell- and context-specific signaling of a complex containing FXII-uPAR and different integrin heterodimers.

This aspect is now addressed in the results section and the discussion:

“To identify the relevant integrin α -subunit interacting with integrin β 1 upon FXII-uPAR stimulation in DKD, we focused on integrins α 6 and α 5, which were the most downregulated alpha subunits (highest FDR values next to integrin β 1) in hyperglycemic *F12*^{-/-} mice kidneys compared to WT mice (Fig. 8a). To determine a possible role of integrins α 6 and α 5, we exposed HKC-8 cells to FXII and performed coimmunoprecipitation. While exposure of HKC-8 cells to FXII increased the integrin β 1/ α 6 interaction, the integrin β 1/ α 5 interaction was not affected (Fig. S19a). To scrutinize whether the integrin α 6 β 1 heterodimer mediates FXII-uPAR-dependent tubular injury and senescence, we blocked integrin α 6 with a functional blocking monoclonal antibody (1 μ g/ml). Blocking integrin α 6 abolished FXII-induced induction of injury markers in HKC-8 cells (Fig. S19b,c).”

(page 8, line 34-42)

and in the discussion:

“Our results showed (i) that the uPAR/FXII complex required active integrin β 1 but not β 3, that (ii) a blocking antibody against integrin β 1 almost completely prevented FXII-induced tubular cell injury, that (iii) the integrin β 1 forms a heterodimer with integrin α 6, which promotes FXII-associated DNA damage and tubular cell injury. The integrin α 6 β 1 heterodimer induces ROS, DNA damage and senescence in human fibroblasts ⁹⁴, similar to the phenotype we observed in kidney tubular cells in our study.”

(page 12, line 32-37)

Fig. R1-1 (a) Volcano plot comparing integrins expression (green) in hyperglycemic *F12*^{-/-} mice to hyperglycemic WT mice based on Log fold change (FC) values and false discovery rate (FDR). Integrin $\beta 1$ (*Itgb1*), $\alpha 6$ (*Itga6*), and $\alpha 5$ (*Itga5*) are the most downregulated integrins in hyperglycemic *F12*^{-/-} mice kidneys by FDR. **(b)** Representative immunoblots for integrin $\alpha 6$ and integrin $\alpha 5$ from integrin $\beta 1$ coimmunoprecipitation (IP, top) and immunoblots for integrin $\beta 1$ and α -Tubulin from the input (input, bottom,) of HKC-8 cells exposed to purified human FXII (62nM) in the presence of Zn²⁺ (10 μ M) for 24 h (FXII) compared to control non-treated cells (C). Input serves as a loading control. **(c,d)** Representative immunoblots (c, loading control: α -Tubulin) and dot plot summarizing results (d) for γ -H2AX, and KIM-1 expression in HKC-8 cells exposed to purified human FXII (62nM) in the presence of Zn²⁺ (10 μ M) for 24 h with and without pretreatment with a monoclonal integrin $\alpha 6$ function blocking antibody (1 μ g/ml) for 30 min. Dot plots reflecting mean \pm SEM of 3 independent experiments; one-way ANOVA with Tukeys’s multiple comparison test. * $P < 0.05$, ** $P < 0.01$, *** $P < 0.001$.

Comment 1-2:

Are FXII levels of 62 nM achievable in proximal tubule in diabetic nephropathy? Teleologically, why would PTCs have a pathway for injury if the agonist is not physiologically present? What is the physiologic role of FXII in the proximal tubule as you understand it from these elegant studies?

Response 1-2:

We thank the reviewer for this important comment. The physiological role of FXII is generally not well understood. FXII-deficient individuals are phenotypically normal and do not show excessive bleeding risks¹¹. In the current study we propose a new function of FXII in the kidney where FXII appears to be secreted from tubular cells and to signal in an auto- or a paracrine fashion. Whether FXII secreted from tubular cells has a physiological function remains unknown. Based on our current findings we hypothesize that FXII released from the tubular cells provides a danger signal. We speculate that short term injury and short term FXII release initiates tissue healing, while persistent FXII secretion and prolonged stimulation of the signaling cascade under chronic disease conditions, as in DKD, promotes pathological consequences such as senescence. In line with this hypothesis, analysis of publicly available single cell transcriptomic dataset (Kidney Interactive Transcriptomics; KIT) of mouse kidney ischemia reperfusion injury (IRI) elucidates a short-term induction of *F12* expression in tubular cells. In kidney IRI (acute injury), tubular *F12* is transiently

upregulated at early stages (within 2 days post injury), while in other kidney cell types FXII induction shows a delayed induction¹². On the other hand, in unilateral ureteral obstruction, a model of persistent kidney injury, high *F12* expression is observed in some tubular cells at 14 days post injury¹³. Another established example of a dual effect of a protein in acute versus chronic renal injury is the cell cycle inhibitor p21: induction of p21 in acute injury is protective, while its persistent induction is harmful¹⁴⁻¹⁶. Whether FXII has a similar dual function needs to be addressed in future studies.

The concentrations reached on the tubular cell surface are not expected to be reflected by what is measured in the urine, as FXII may be taken up upon receptor binding and as protein concentrations generally change strongly along the tubular system. Hence, the local concentration of FXII on tubular cells, in particular if secreted locally, is difficult to predict.

This point is now emphasized in the discussion section:

“The precise role of tubular FXII in kidney injury, e.g. a differential effect in acute versus chronic renal injury, requires further investigation, including, for example, mouse models with cell-specific FXII-inactivation.”

(Page 11, line 31-32)

and

“In addition, while our data propose that urinary FXII reflects tubular injury, the levels measured in urine most likely do not reflect levels reached locally in the tubular compartment or at the surface of tubular cells. Further studies are required to elucidate the exact role and the regulation of tubular FXII in the pathogenesis of DKD.”

(Page 11, line 40-43)

Comment 1-3:

Although in fact I agree with the authors' finding that some FXII signaling pathway is mediated through uPAR and beta--1-integrin based on what is known with previously published work on endothelial cells, most investigators will not think it is convincing with just one inhibitory experiment with a peptide (PGS20) from domain 2 of uPAR.

Response 1-3:

We appreciate this comment. We concur with the reviewer that further evidence is required to support the involvement of uPAR in FXII signaling on renal tubular cells. Based on our computational modeling, we have identified candidate interacting residues in different amino acid stretches in domains 1 and 2 of uPAR that interact with corresponding residues from fibronectin type II (FN2) and Kringle domains of the heavy chain of FXII. To validate the relevance of these identified uPAR residues for the interaction, we have synthesized 3 uPAR-based peptides covering these identified amino acid stretches, namely RL20 (⁵²RLWEEGEELELVEKSC⁷¹HSE⁷¹) and DL19 (⁹⁶DLCNQGNSGRAV¹¹⁴TYSRSRY¹¹⁴) from domain 1 in addition to DV20 (¹⁴⁶DVVTHW¹⁶⁵IQEGEEGRPKDDR¹⁶⁵H¹⁶⁵) from domain 2. The peptide DV20 shares some sequence similarity with the previously reported peptide IQE13, a peptide from uPAR domain 2 that binds integrins, and shares a few amino acids with the previously reported peptides QCR20 and EEG20 from uPAR domain 2¹⁷. We used Alphafold2 multimer to predict binding of these newly synthesized peptides to the heavy chain of FXII. Global docking of the peptides using Alphafold2 multimer predicted binding to the FXII interface for the 3 peptides (**Fig. R1-3a**).

To experimentally validate the relevance of these peptides for the interaction, we pretreated HKC-8 cells with the peptides RL20, DL19, and DV20 (300 μM). DL19 and DV20 reduced FXII binding to uPAR as determined by coimmunoprecipitation (**Fig. R1-3b**). Pretreatment with the peptides DL19 and DV20 reduced FXII-induced DNA damage and senescence markers (**Fig. R1-3b,c**). These newly generated results in addition to the results we showed in the original manuscript with the peptide PGS20 indicate that FXII interacts with uPAR domain 2 at multiple sites on tubular cells. Furthermore, the blocking effects of the peptide DL19 from uPAR domain 1 indicates a new binding site on this uPAR domain for FXII that has – to the best of our knowledge – not been described previously. The area covered by the peptide DL19 has been reported previously to interfere with the binding of high molecular weight kininogen to uPAR¹⁸. Binding of the Kringle domain of FXII to domain 1 of uPAR seems possible, taken into account the following evidence: (i) the important role of the Kringle domain of urokinase-type plasminogen activator (uPA) to uPAR binding¹⁹, (ii) the inhibition of this binding by peptides targeting uPAR domain 1²⁰, and the high sequence similarity between the Kringle domains of FXII and uPA. Further studies are required to characterize these newly synthesized peptides in different cells and in cell-free systems to validate our results. Furthermore, structural modifications to these peptides may help to increase their affinity and stability for future studies incorporating them as therapeutic options *in vivo*. This point is now addressed in the results section:

“Furthermore, we have synthesized 3 uPAR-based peptides based on the identified amino acids required for the uPAR-FXII interaction, namely RL20 (⁵²RLWEEGEELELVEKSC⁷¹HSE⁷¹) and DL19 (⁹⁶DLCNQGNSGRAV¹¹⁴TYSRSRY¹¹⁴) from domain 1 and DV20 (¹⁴⁶DVVTHW¹⁶⁵IQEGEEGRPKDDR¹⁶⁵H¹⁶⁵) from domain 2 (Table S5). The area covered by the peptide DL19 has been reported previously to interfere with the binding of HK to uPAR⁴⁵. Additionally, the peptide DV20 shares some sequence similarity with the previously reported peptide IQE13, a peptide from uPAR domain 2 that binds integrins, in addition to sharing few amino acid sequences with the previously reported peptides

QCR20 and EEG20 from uPAR domain 2³. We used AlphaFold2_multimer_v3 to predict the binding of these newly synthesized peptides to the heavy chain of FXII. Global docking of the peptides using AlphaFold2_multimer_v3 predicted binding to the FXII interface for the 3 peptides (Fig. S12f). Pretreatment of HKC-8 cells with the peptides DL19 and DV20 (300 μM) reduced FXII binding to uPAR as determined by coimmunoprecipitation, while RL20 had no effect in our cellular model (Fig. 7d). Pretreatment with the peptides DL19 and DV20 reduced the induction of DNA damage and senescence markers by FXII or by the combination of FXII-based peptides HR13 and PW15, suggesting a functional relevance of the amino acid stretches D96-Y114 and D146-H165 in uPAR domains 1 and 2, respectively, for FXII-induced uPAR signaling (Fig. 7e; Fig. S12g-i).”

(Page 7, line 25-39)

and

“This modeling and our experimental data identify new molecular interactions of FXII and uPAR, extending previous findings³. Furthermore, the blocking effects of the peptides DL19 and PW15 derived from uPAR domain 1 and FXII Kringle domains, respectively, suggest new binding sites of the FXII-uPAR interaction. Previous studies suggested the involvement of FXII’s Kringle domain in binding to artificial surfaces⁶⁷. Binding of the Kringle domain of FXII to domains 1 and 2 of uPAR seems possible, taken into account the following evidence: (i) the important role of the Kringle domain of urokinase-type plasminogen activator (uPA) to uPAR binding⁶⁸, (ii) the high sequence similarity between the Kringle domains of FXII and uPA, and (iii) the inhibition of this binding by peptides targeting uPAR domains 1 and 2⁶⁹. Our computational docking predicted binding of the synthesized peptides to their corresponding residues, yet not all of them showed inhibitory effects on the FXII-uPAR interaction in our experiments. These differences may reflect discrepancies between the in silico and in vitro approach or indicate that the interacting residues between FXII and uPAR depend on the specific cell type and experimental conditions. Interestingly, the experimental results indicate that the simultaneous interaction of FXII with different uPAR binding sites is required for signal transduction and induction of DNA damage and senescence. Accordingly, blocking one binding site is sufficient to inhibit FXII’s effect. Further detailed analyses of the FXII-uPAR interaction may hence identify new molecular targets allowing to therapeutically modulate this interaction.”

(page 11, line 2-18)

Fig. R1-3 (a) Representative images of the ALphaFold2 prediction of peptides binding to the heavy chain of FXII. (b) Representative immunoblots for FXII and uPAR from uPAR coimmunoprecipitation (IP, top) and immunoblots for uPAR, and α -Tubulin from the input (input, bottom) of HKC-8 cells exposed to purified human FXII (62nM) in the presence of Zn²⁺ (10 μ M) for 24 h with and without pretreatment with uPAR based peptides (RL20, DL19, and DV20; 300 μ M) for 1 h compared to control non-treated cells. Input serves as a loading control. (c,d) Representative immunoblots (c, loading control: α -Tubulin) and dot plot summarizing results (d) for γ -H2AX, p21, and KIM-1 expression in HKC-8 cells exposed to purified human FXII (62nM) in the presence of Zn²⁺ (10 μ M) for 24 h with and without pretreatment with uPAR-based peptides (DL19 and DV20; 300 μ M) for 1 h compared to control non-treated cells. Dot plots reflecting mean \pm SEM of 3 independent experiments; one-way ANOVA with Tukeys's multiple comparison test. * P <0.05, ** P <0.01, *** P <0.001, ns: non-significant.

Comment 1-4:

However, in glomerular-tubular fluid from a leaky glomerulus in a diabetic with Kimmelstiel-Wilson glomerulosclerosis, the plasma proteins high molecular weight kininogen, low molecular weight kininogen, vitronectin, and single chain urokinase may all be present. If so and in molar excess, can they block FXII-induced renal tubular cell toxicity inducing oxidative changes and DNA damage?

Response 1-4:

We appreciate this interesting comment. We agree with the reviewer that the glomerular filtration barrier is disrupted in particular at advanced DKD stages at which the characteristic Kimmelstiel-Wilson lesions are observed, leading to the leakage of proteins that may compete with FXII binding to uPAR. Kimmelstiel-Wilson lesions are not observed in mouse models of DKD and hence we aimed to detect early changes associated with DKD in our study. This has the advantage that any biomarker or mechanism observed reflects early disease stages of DKD, which are in principle therapeutically amendable. Indeed, in our patient cohort we observed a slight and significant increase of FXII already at the lowest CKD-risk category (Fig. 1e). Based on the current results, we hypothesize that FXII is upregulated and excreted by renal

tubular cells which contributes in a para- or autocrine fashion, by signaling through uPAR, to the induction of oxidative DNA damage and senescence at early disease stages. In addition, comparison of the gene expression levels of *F12*, *KNG1* (encoding HK), and *VTN* (encoding vitronectin) in the tubulointerstitial compartment in 2 different publicly available datasets indicates upregulation of *F12*, downregulation of *KNG1*, while *VTN* expression did not show a significant change (**Fig. R1-4a,b**). Kidneys of diabetic mice showed a similar phenotype of *F12* upregulation and *Kng1* downregulation as detected by qPCR (**Fig. R1-4c**). In human DKD, we observed upregulation and colocalization of FXII and uPAR in DKD biopsies from patients at advanced disease stages compared to controls, while HK was barely upregulated and did not show colocalization with uPAR in DKD biopsies (**Fig. R1-4d-h**), supporting the notion that HK is unlikely to compete with FXII in the tubular compartment.

To experimentally address the effect of excess molar concentrations of other competing proteins to the binding of FXII to uPAR (as expected at advanced disease stages), we exposed HKC-8 cells to FXII (62 nM; in the presence and absence of zinc, 10 μ M), in the presence of equimolar (62 nM) or excess molar (120 nM) concentrations of HK and determined FXII binding to uPAR by coimmunoprecipitation. The presence of equimolar concentration of HK did not affect FXII binding to uPAR in the presence of zinc, while excess molar concentration of HK reduced the binding even in the presence of zinc (**Fig. R1-4i**).

Together these data support a model in which the local expression of FXII in renal tubular cells in early DKD initiates uPAR signaling and thus promotes DNA damage. We cannot exclude that at advanced stages, excess molar concentrations of competing proteins such as HK may modulate the FXII-uPAR interaction and signaling. Whether this has an impact on the already established tubular senescence and injury remains to be studied in the future. This comment is addressed in the results section:

“Both assays revealed an increase in the FXII-uPAR interaction in high glucose conditions (Fig. 6e,f; Fig. S10c). Furthermore, increased interaction of FXII and uPAR was detected in human DKD biopsies compared to control biopsies using PLA (Fig. 6g,h). To investigate whether other plasma proteins that compete with FXII to uPAR binding such as high molecular weight kininogen (HK) and vitronectin² may interfere with FXII-uPAR binding, we analyzed the Nephroseq[®] and the Karokidney transcriptomic databases. While tubular *F12* expression was increased in DKD, the expressions of *KNG1* (encoding HK) was downregulated and *VTN* (encoding vitronectin) was not changed compared to controls (Fig. S11a,b). Analysis of FXII, HK and uPAR expression in human DKD biopsies revealed upregulation and colocalization of FXII and uPAR, while HK signal was barely detectable and showed little if any colocalization with the upregulated uPAR (Fig. S11c-g). To experimentally address the effects of competing proteins on FXII-uPAR binding on renal tubular cells, we exposed HKC-8 cells to FXII (62 nM; \pm 10 μ M zinc) in the presence of equimolar (62 nM) or excess molar (120 nM) concentrations of HK and determined FXII binding to uPAR by coimmunoprecipitation. The presence of equimolar concentration of HK did not affect FXII binding to uPAR in the presence of zinc, while excess molar concentration of HK reduced the binding even in the presence of zinc (Fig. S11h). Taken together, these data support a model in which the local expression of FXII in renal tubular cells in early DKD initiates uPAR signaling and thus promotes DNA damage.”

(page 6, line 34 - page 7 line 2)

and in the discussion:

“Expression profiling revealed that the protection against kidney injury in hyperglycemic *F12*^{-/-} mice was linked to negative enrichment of key pathological pathways related to DKD, including cell cycle arrest. Based on the mouse model used and the clinical data, we speculate that FXII-uPAR signaling is an early event in the course of DKD leading to tubular ROS and senescence. We cannot exclude that at advanced stages excess molar concentrations of competing proteins such as HK may cross the dysfunctional glomerular filtration barrier and modulate FXII-uPAR signaling. Whether this would have an impact on already established tubular senescence remains to be studied.”

(page 10, line 23-30)

Fig. R1-4 (a,b) Heatmaps summarizing *F12*, *KNG1*, and *VTN* expression in the tubulointerstitial compartment comparing diabetic kidney disease patients (DKD) to non-diabetic control healthy donors (C) in Karokidney (a) and Nephroseq (b) public databases. Unpaired student's t-test. **(c)** Dot plots representing the expression of *F12* and *Kng1* expression in normoglycemic and hyperglycemic WT mice kidneys (qPCR). Dot plots reflecting mean \pm SEM of 4 mice per group; unpaired student's t-test, *P<0.05, ***P<0.001. **(d-g)** Representative histological images of FXII (red) and uPAR (green; d) or HK (red) and uPAR (green; f) in human kidney sections of nondiabetic controls (C) or diabetic patients with DKD (DKD); DAPI nuclear counterstain, blue. Scale bars represent 20 μ m, and dot plots summarizing the colocalization intensity to the uPAR intensity in both conditions (e,g). Dot plots reflecting mean \pm SEM; unpaired student's t-test. ***P<0.001, ns: non-significant. **(h)** Magnified areas of the stainings (represented by the white squares in d and f) and histograms showing the intensities of the of FXII (red)-uPAR (green) or HK (red)-uPAR (green) across the drawn solid lines represented in the magnified images. **(i)** Representative immunoblots for FXII and uPAR from uPAR coimmunoprecipitation (IP, top) and immunoblots for uPAR, and α -Tubulin from the input (input, bottom) of HKC-8 cells exposed to purified human FXII (62 nM); in the presence and absence of zinc, 10 μ M, in the presence of equimolar (62 nM) or excess molar (120 nM) concentrations of HK. Input serves as a loading control.

Comment 1-5:

Also, does FXII induce apoptosis in renal tubular cells? Apparently, apoptosis is not an entity in biologic senescence.

Response 1-5:

We thank the reviewer for this question. Senescent cells upregulate anti-apoptotic proteins which contribute to their resistance to apoptosis^{21,22}. Accordingly, one would not expect that FXII-mediated senescence induces apoptosis. To address this question, we determined levels of the antiapoptotic proteins BCL-2 and BCL-XL in kidney lysates of normoglycemic and hyperglycemic WT and *F12*^{-/-} mice. Both proteins were upregulated in hyperglycemic WT kidneys consistent with increased senescence compared to hyperglycemic *F12*^{-/-} mice (**Fig. R1-5a,b**). Additionally, we determined the extent of apoptosis in kidneys of WT and *F12*^{-/-} mice by cleaved caspase-3 immunostaining and did not observe a difference (**Fig. R1-5c,d**). This point is now addressed in the results section:

“Senescent cells induce anti-apoptotic regulators and are not hallmarked by increased apoptosis^{38,39}. The anti-apoptotic regulators BCL-2 and BCL-XL were induced in hyperglycemic WT compared to *F12*^{-/-} mice kidneys and the number of cleaved caspase-3 positive cells was not different in the kidneys of both genotypes (Fig. S7). These findings are consistent with increased kidney senescence in WT, but not *F12*^{-/-} mice. Thus, loss of FXII expression protects mice from senescence and inflammation in DKD”

(page 5, line 41-45)

Fig. R1-5 (a,b) Representative immunoblots (a, loading control: α -Tubulin) and dot plot summarizing results (b) for BCL-2 and BCL-XL expression in kidney lysates of normoglycemic controls (C) and hyperglycemic (DM) wild type (WT) and *F12*^{-/-} mice. Dot plot reflecting mean \pm SEM of 4 mice per group; two-way ANOVA with Tukey's multiple comparison test. * P <0.05, *** P <0.001, ns: non-significant. **(c,d)** Exemplary histological images (c) and dot plot summarizing results (d) of the cleaved Caspase-3 (clv. Casp-3) positive cells in experimental groups (as described in a). clv. Casp-3 is immunofluorescently detected, green; DAPI nuclear counterstain, blue. Scale bars represent 20 μ m. Dot plot reflecting mean \pm SEM of 5 mice per group; unpaired student's t-test, ns: non-significant.

Comment 1-6:

Last, I noticed that you used an older version of AlphaFold. Presently, AlphaFold2-multimer is an improvement on the original program. It would be of interest to see if the original data holds up and there are additional interactions observed.

Response 1-6:

We appreciate this comment. We would like to kindly point out to the reviewer that we actually used the latest version AlphaFold2_multimer_v3. The information, which we provided in the supplementary methods section, stated that we used "AlphaFold2", but did not specify the version. This misunderstanding is now clarified in the results section:

"To identify the interacting residues of FXII and uPAR, we performed computational modeling of the heavy chain of FXII and uPAR using AlphaFold2_multimer_v3 (see supplementary methods)"

(page 7, line 5-6)

Specific comments.

Introduction.

Comment 1-7:

p.3, line 4 from the top. How does one define "senescence" in biologic terms? Is the reduction of leukocyte presence and activity a hallmark of biologic senescence?

Response 1-7:

We thank the reviewer for this comment. SASP factors promote immune cell infiltration, presumably in an attempt to initially remove senescent cells from the affected tissues. However, with the persistence of senescence-inducing stimuli and an associated proinflammatory microenvironment, this process can become inefficient leading to the accumulation of senescent and inflammatory cells, contributing to the overall pathology of DKD ²³.

Biological senescence is characterized by the following hallmarks:

- 1. Cell cycle arrest:** which we identified by the pathways from the RNA-seq data, and confirmed by immunostaining for p21, Ki-67, and the cell cycle analysis in tubular cells.
- 2. Macromolecular damage:** which we identified by immunostaining for the DNA damage markers (γ -H2AX & 8-O-dG), SAHF, and the loss of lamin B1 indicating nuclear membrane instability.
- 3. Morphological changes,** which we identified by hypertrophy reflected by kidney/body weight ratio and the increased activity of the β -gal staining.
- 4. The secretory phenotype (SASP):** which we have identified in the RNA-seq data and confirmed by qPCR and Olink assay. SASP factors contribute to inflammation by stimulating the recruitment of immune cells, in addition to promoting fibrosis and ECM rearrangement resulting in organ dysfunction.

So regarding the reviewer's statement, increased infiltration and activity of leucocytes is a hallmark of senescence and not their reduction. Accordingly, we observed a higher infiltration of macrophages (F4/80

immunostaining) in the kidney interstitial space of hyperglycemic WT compared to hyperglycemic *F12*^{-/-} mice (Fig. S6g). To address this point we revised the introduction adding more details describing the senescence phenotype:

“Cellular stressors, such as increased reactive oxygen species (ROS) generation in diabetic kidneys trigger DNA damage and senescence, which is characterized by permanent cell cycle arrest, macromolecular damage, morphological changes, and a specific secretome (SASP, senescence associated secretory phenotype) that induces inflammatory and fibrotic changes and compromises kidney function ^{10,11}.”

(Page 3, line 12-16)

Comment 1-8:

p.3 line 12 from the top. You cite review articles by authors who mostly did not do the work. The citations that first outlined this pathway were Refs #35, 36, 18, and 60.

Response 1-8:

We thank the reviewer for bringing this point to our attention. We corrected the citations to mention the original work based on the reviewer’s suggestion.

“Furthermore, FXII zymogen signals through plasma membrane receptors such as the urokinase-type plasminogen activator receptor (uPAR) in different cells promoting cell and context specific responses including angiogenic effects in endothelial cells, activation of immune cells such as neutrophils and macrophages, and profibrotic effects in lung fibroblasts ²⁻⁶”

(Page 3, line 4-8)

Comment 1-9:

The Introduction overall rambles, i.e., does not show focus on the topic at hand. If this pathway was truly recognized by you in an unbiased way, your later review of the already published FXII literature on the function of zymogen FXII on endothelium, neutrophils, and lung fibroblasts helped confirm your assessment. Your introduction should communicate that knowledge already in the literature.

Response 1-9:

We appreciate this insightful comment by the reviewer. Based on the reviewer’s suggestion, we have rewritten the introduction in parts discussing the previous literature supporting the role of zymogen FXII in different pathologies.

The introduction is written in the revised manuscript as the following:

“Coagulation factor XII (FXII, gene F12) is activated upon interaction with negatively charged surfaces (contact activation). The activated protease (FXIIa) initiates the intrinsic coagulation pathway and inflammatory reactions via the kallikrein-kinin-system (KKS) ¹. Furthermore, FXII zymogen signals through plasma membrane receptors such as the urokinase-type plasminogen activator receptor (uPAR) in different cells promoting cell and context specific responses including angiogenic effects in endothelial cells, activation of immune cells such as neutrophils and macrophages, and profibrotic effects in lung fibroblasts ²⁻⁶.”

Proinflammatory and profibrotic signaling is a hallmark of diabetic kidney disease (DKD), a serious microvascular complication in patients with diabetes mellitus ^{7,8}. The pathomechanisms underlying proinflammatory and profibrotic signaling in DKD involve hemodynamic and metabolic changes as well as DNA damage and senescence ^{9,10}. Cellular stressors, such as increased reactive oxygen species (ROS) generation in diabetic kidneys trigger DNA damage and senescence, which is characterized by permanent cell cycle arrest, macromolecular damage, morphological changes, and a specific secretome (SASP, senescence associated secretory phenotype) that induces inflammatory and fibrotic changes and compromises kidney function ^{10,11}.

Additionally, diabetes mellitus in general and DKD in particular are linked with alterations of the coagulation system that predispose to inflammatory and fibrotic changes ^{12,13}. Despite its known role in thrombosis and bradykinin-driven inflammation, the role of FXII in the pathophysiology of DKD is not yet defined. Earlier reports showed upregulation of hepatic FXII production in patients with insulin resistance and increased levels of circulating FXII and FXIIa in patients with diabetes mellitus or chronic kidney disease (CKD) ¹⁴⁻¹⁷. Neutrophil-derived FXII activates neutrophils in an autocrine fashion, demonstrating that non-hepatic FXII promotes inflammation ⁴, a key feature of DKD.

FXII signaling via uPAR involves uPAR coreceptors such as integrins, promoting cell-specific responses. Interference with FXII-uPAR binding can inhibit FXII-associated signaling effects ^{3,4}. uPAR is associated with senescence and targeting uPAR with CAR-T cells eliminates senescent cells and associated pathologies ¹⁸. Furthermore, a role of uPAR in renal diseases including DKD is established ^{19,20}.

Whether FXII induction in DKD contributes to the inflammatory state through the activation of the intrinsic coagulation pathway, the KKS or is mechanistically linked with DKD through a signaling mechanism independent of its protease function remains unknown. It is possible but remains to be shown that interaction of FXII with uPAR promotes senescence in diabetic kidneys. Deciphering the relevance of FXII binding to uPAR for induction of senescence may be therapeutically relevant, as strategies inhibiting zymogen FXII or its active form do not increase the risk of bleeding and are hence considered safe ²¹.

In the current study, we combined unbiased approaches, analyses of diabetes patient samples and murine diabetes models to uncover a function of zymogen FXII signaling for DKD pathology, which promotes oxidative DNA damage and tubular senescence via uPAR-integrin β 1 signaling.”

(page 3)

Methods.

Comment 1-10:

S4. p.11, lines 26, 27. What are the sequences for FXII peptides HR13, TY10, and PW15? No citation is provided in the manuscript text. The Supplementary table citation should be named in the main text. The original source of the peptides should be named, as well. Were they completed intuited from alpha fold, or did you see them in other literature sources as well? HR13 had previously been reported as an artificial surface binding site by Citarella et al. 2000 and shown to block FXII binding to HUVEC by Mahdi et al. 2002. Has PW15 been characterized before your alpha fold studies?

Response 1-10:

We thank the reviewer for this comment. The three used peptides were identified based on the interacting residues predicted by AlphaFold2-multimer-v3 simulation results which were in line with previously published data. We agree with the reviewer that the area covered by the peptide HR13 on FN2 domain of FXII has been identified by Citarella et al. to be involved in binding negatively charged surfaces²⁴. Furthermore, HR13 shares sequence similarity in 8 amino acids out of 13 with the previously published peptide YHK9 which has been shown to block FXII binding to HUVEC cell by Mahdi et al.²⁵.

Regarding the peptides TY10 and PW15 from the Kringle domain of FXII, these peptides have not been characterized before to the best of our knowledge. Previous studies suggested the involvement of the Kringle domain in binding of FXII to artificial surfaces²⁶. A monoclonal antibody directed against the Kringle domain (MoAb F1) did not interfere with FXII binding to negatively charged surfaces, while it increased its activation in the plasma^{24,27}.

To address this comment, we included the full sequence of the peptides in the main text in addition to including the citations for HR13. In the results section we wrote:

“To confirm the importance of the identified FXII residues, we designed sequential peptides covering these residues on the FN2 (HR13: ⁵⁴HRQLYHKCTHKGR⁶⁶) and Kringle (TY10: ²⁴⁶TYRNVTAEQA²⁵⁵, PW15: ²⁷⁵PWCFVLNRDRLSWEY²⁸⁹) domains (Table S5). The area covered by the peptide HR13 derived from the FN2 domain of FXII has been previously shown to mediate binding to negatively charged surfaces⁴⁴. Furthermore, HR13 shares sequence similarity in 8 amino acids out of 13 with the previously published peptide YHK9 which blocks FXII binding to HUVEC cells².”

(page 7, line 8-14)

and in the discussion we state:

“This modeling and our experimental data identify new molecular interactions of FXII and uPAR, extending previous findings³. Furthermore, the blocking effects of the peptides DL19 and PW15 derived from uPAR domain 1 and FXII Kringle domains, respectively, suggest new binding sites of the FXII-uPAR interaction. Previous studies suggested the involvement of FXII’s Kringle domain in binding to artificial surfaces⁶⁷. Binding of the Kringle domain of FXII to domains 1 and 2 of uPAR seems possible, taken into account the following evidence: (i) the important role of the Kringle domain of urokinase-type plasminogen activator (uPA) to uPAR binding⁶⁸, (ii) the high sequence similarity between the Kringle domains of FXII and uPA, and (iii) the inhibition of this binding by peptides targeting uPAR domains 1 and 2⁶⁹. Our computational docking predicted binding of the synthesized peptides to their corresponding residues, yet not all of them showed inhibitory effects on the FXII-uPAR interaction in our experiments. These differences may reflect discrepancies between the in silico and in vitro approach or indicate that the interacting residues between FXII and uPAR depend on the specific cell type and experimental conditions.”

(page 11, line 2-14)

Comment 1-11:

p.11, line 29. What is the citation or source of FXII cyclic peptide inhibitors; FXII-618, FXII-900? Again, the Supplementary table citation should be named in the METHODS. The original source should be named and put in the main references of the paper, as well.

Response 1-11:

We appreciate this comment by the reviewer. We would like to kindly point out that references citing the cyclic inhibitors were mentioned in the main text. We agree with the reviewer that the references were missing from the supplementary table. Based on the reviewer's suggestion, we included now the references in the supplementary methods and tables in addition to the main text. In the results section we wrote:

“To determine whether the observed effects require FXII's proteolytic activity, we used two different approaches: i) by using the FXII cyclic peptide inhibitors FXII-618 and FXII-900^{47,48} (Table S5)”

(page 7, line 47-48)

Results.

Comment 1-12:

p.5, lines 10,11. Collectively, it looks like the absence of FXII upregulates gene sets that could be related to diabetes and downregulates gene sets related to inflammation and cell function, i.e., slowing processes down. Your statement is not this clear.

Response 1-12:

We thank the reviewer for this interesting comment. Deficiency of FXII was associated with upregulation of metabolic pathways including lipid metabolism, amino acid metabolism and organic acid metabolism compared to WT mice. Renal tubular cells largely rely on fatty acid oxidation and utilization of amino acids in the process of gluconeogenesis and kidney homeostasis^{28,29}. The downregulation of these metabolic pathways under DKD reflects tubular dysfunction as observed in WT mice kidneys, while the upregulation of these pathways in *F12*^{-/-} mice reflects a protective phenotype. On the other hand, FXII deficiency downregulates pathways related to cell cycle arrest, inflammation and fibrosis compared to WT mice. Based on the reviewer's suggestion, we rewrote this part to better describe the changes observed in our RNA seq profiling data as the following:

“Further pathway analysis of the differentially expressed genes (DEGs) indicated that FXII deficiency was associated with downregulation of pathways involved in cell cycle regulation, cell adhesion, integrin signaling, inflammation, and hemostasis (Fig. 3d; Fig. S3b-d). On the other hand, compared to WT mice, *F12* deficiency (i) upregulated DNA damage repair pathways and (ii) upregulated metabolic pathways such as lipid metabolism, amino acid metabolism, and organic acid metabolism^{24,25} (Fig. S4). Collectively, FXII regulates gene sets related to pathways linked to DKD in hyperglycemic kidneys^{12,26-28}.”

(page 5, line 7-13)

Comment 1-13:

p.5, lines 25,26. What is the source of the senescence gene set? Where is this group of genes or gene set derived?

Response 1-13:

The thank the reviewer for this comment. This list is derived from previous publications showing gene sets associated with senescence in mouse kidneys^{30,31} and from a commercially available panel of mouse senescence genes (Qiagen, Cat. no. 330231 PAMM-050ZA). In the revised results section, we write:

“Furthermore, we analyzed a panel of genes known to be associated with senescence in mice^{35,36}. These genes were downregulated in the kidneys of hyperglycemic *F12*^{-/-} compared to WT mice (Fig. 4d).”

(page 5, line 27-29)

Comment 1-14:

Supplementary Figure S9d. Does peptide HR13, PW15, or combined HR13-PW15 alone stimulated gamma-H2AX, p21, or KIM-1 increase? This experiment is an appropriate control to the figure as presented. If so, is the pathway blocked uPAR peptide PGS20 and the peptide on beta-1-integrin (IQE13 - see LaRusch et. al.) that binds uPAR's domain 1?

Response 1-14:

We appreciate this interesting comment by the reviewer. To experimentally address this point, we exposed HKC-8 cells to peptides HR13, PW15 or a combination of both (300 μM). Interestingly, exposure to single peptides did not induce γ-H2AX, p21, or KIM-1, while the combination of both peptides induced these markers similar to FXII (**Fig. R1-14a,b**). To investigate whether our newly tested uPAR-based peptides can block the effects elicited by the combination of HR13 and PW15, we pretreated the cells with the peptide DL19 (uPAR domain 1) and DV20 (uPAR domain 2, each 300 μM) followed by exposure to combined HR13/PW15 (300 μM). The peptide DV20 shares some sequence similarity with the peptide IQE13 as we described in point 1.3. Pretreatment with the peptides DL19 and DV20 ameliorated the induction of injury and senescence markers (**Fig. R1-14c,d**).

These results suggest that the simultaneous interaction of FXII with different uPAR binding sites is required for signal transduction and induction of DNA damage and senescence. Accordingly, blocking one binding site is sufficient to inhibit FXII's effect. This point is addressed in the revised results section:

“Exposure of HKC-8 cells simultaneously to the peptides HR13 and PW15 induced markers of DNA damage and senescence, while single peptides (HR13 or PW15) failed to induce this response (Fig. S12d,e). These results suggest that the simultaneous interaction of FXII with different uPAR binding sites is required for signal transduction and induction of DNA damage and senescence, and that a combination of FXII-derived peptides can mimic the effect.”

(page 7, line 20-24)

and

“Pretreatment with the peptides DL19 and DV20 reduced the induction of DNA damage and senescence markers by FXII or by the combination of FXII-based peptides HR13 and PW15, suggesting a functional relevance of the amino acid stretches D96-Y114 and D146-H165 in uPAR domains 1 and 2, respectively, for FXII-induced uPAR signaling (Fig. 7e; Fig. S12g-i).”

(page 7, line 36-39)

and in the discussion:

“Interestingly, the experimental results indicate that the simultaneous interaction of FXII with different uPAR binding sites is required for signal transduction and induction of DNA damage and senescence. Accordingly, blocking one binding site is sufficient to inhibit FXII’s effect. Further detailed analyses of the FXII-uPAR interaction may hence identify new molecular targets allowing to therapeutically modulate this interaction.”

(page 11, line 14-18)

Fig. R1-14 (a,b) Representative immunoblots (a, loading control: α -Tubulin) and dot plot summarizing results (c) for γ -H2AX, p21, and KIM-1 expression in HKC-8 cells peptides HR13, PW15 or a combination of both (300 μ M) in the presence of Zn²⁺ (10 μ M) for 24 h compared to control non-treated cells. Dot plots reflecting mean \pm SEM of 3 independent experiments; one-way ANOVA with Tukeys’s multiple comparison test. * P <0.05, ** P <0.01, *** P <0.001, ns: non-significant. **(c,d)** Representative immunoblots (c, loading control: α -Tubulin) and dot plot summarizing results (d) for γ -H2AX, p21, and KIM-1 expression in HKC-8 cells exposed to a combination of peptides HR13 and PW15 (300 μ M) in the presence of Zn²⁺ (10 μ M) with and without pretreatment with the uPAR-based peptides DL19 and DV20 (300 μ M) for 1 h compared to control non-treated cells. Dot plots reflecting mean \pm SEM of 3 independent experiments; one-way ANOVA with Tukeys’s multiple comparison test. ** P <0.01, *** P <0.001.

Comment 1-15:

p.7, lines 4-6. Were the authors able to confirm the downregulation of several integrins and the upregulation of beta-1-integrin from their RNA-Seq data, independent and in addition to their PCR data presented in Fig 7b?

Response 1-15:

We thank the reviewer for this comment. We checked the expression of integrins $\beta 1$ and $\alpha 6$ in kidney lysates by immunoblotting and confirmed their downregulation in $F12^{-/-}$ mice compared to WT mice in accordance to our expression profiling data (Fig. R1-15a,b). The volcano plot that we showed in figure 7a (now 8a) represented the regulation of integrins from our RNA seq data (green dots). We only labeled Itgb1 on the volcano blot thus it was not clear if other integrins were down- or upregulated. In order to clarify this point, we labeled the top downregulated integrins in the volcano blot and included a heat map from our RNA seq data summarizing the expression of integrins (Fig. R1-15c,d). The figures are added now in the revised manuscript (Fig. S16).

Fig. R1-15 (a,b) Representative immunoblots (a, loading control: α -Tubulin) and dot plots summarizing results (b) for Itg. $\beta 1$ and $\alpha 6$ expression in kidney lysates of normoglycemic controls (C) and hyperglycemic (DM) wild type (WT) and $F12^{-/-}$ mice. Dot plots reflecting mean \pm SEM of 3 mice per group; two-way ANOVA with Tukey's multiple comparison test. * $P < 0.05$, ** $P < 0.01$, *** $P < 0.001$, ns: non-significant. **(c)** Volcano plot comparing hyperglycemic $F12^{-/-}$ mice to hyperglycemic WT mice based on Log fold change (FC) values and false discovery rate (FDR) of integrins expression. **(d)** Heatmap of the RNA-seq data showing the expression of different integrins in WT-DM and $F12^{-/-}$ -DM mice. Each column represents data from an individual mouse. Color intensity represents row Z-score.

Comment 1-16:

Figure 7i. This very important figure would best be represented by a panel with DAPI alone, uPAR alone, ab Beta-1-integrin alone, and FXII alone and then uPAR-FXII, uPAR-beta-1-integrin, beta-1-integrin-FXII, and then DAPI/PLA (uPAR, beta-1-integrin/FXII). The whole interaction will be visualized.

Response 1-16:

We appreciate this important suggestion by the reviewer. We added the single channels (DAPI, FXII, and PLA of uPAR-integrin $\beta 1$) of the staining presented in figure 7i to better visualize the interaction (**Fig. R1-16a**). Additionally, we conducted a PLA of FXII and uPAR and confirmed the increased interaction of FXII-uPAR in DKD biopsies (**Fig. R1-16b,c**). Furthermore, and in order to better represent the whole interaction, we conducted a triple staining of FXII, uPAR, and integrin $\beta 1$ in human kidney biopsies and observed the upregulation and colocalization of the three proteins in DKD biopsies compared to control biopsies (**Fig. R1-16d**). To address this point in the results section we wrote:

“Strong interaction of uPAR and integrin $\beta 1$ was readily detectable in kidneys of DKD patients, which was accompanied by FXII upregulation (Fig. 8h; Fig. S17c). Furthermore, immunostaining for FXII, uPAR and integrin $\beta 1$ showed colocalization of the three proteins in DKD patient biopsies compared to controls (Fig. S18).”

(page 8, line 30-33)

Fig. R1-16 (a) Representative histological images of proximity ligation assay (PLA, red dots representing uPAR and active integrin $\beta 1$ interaction) in human kidney sections of nondiabetic controls (C) or diabetic patients with DKD (DKD); DAPI nuclear counterstain (blue) and FXII (green). Scale bars represent 20 μm . **(b,c)** Representative histological images of proximity ligation assay (c, PLA, red dots representing FXII and active uPAR interaction) and dot plot summarizing results (d) in human kidney sections of nondiabetic controls (C) or diabetic patients with DKD (DKD); DAPI nuclear counterstain (blue). Scale bars represent 20 μm . Dot plot reflecting mean \pm SEM of 5 samples per group with each dot representing the mean of PLA signals/field for one sample; unpaired student's t-test. *** $P < 0.001$. **(d)** Representative histological images of immunostaining of FXII (white), uPAR (green) and active integrin $\beta 1$ in human kidney sections of nondiabetic controls (C) or diabetic patients with DKD (DKD); DAPI nuclear counterstain, (blue). Scale bars represent 20 μm .

Comment 1-17:

p. 8, lines 42,43. Without question, your work has extended previous work, but as mentioned already above, peptide HR13 had previously been reported as an artificial surface binding site by Citarella et al. 2000 and shown to bind HUVEC by Mahdi et al. 2002.

Response 1-17:

We thank the reviewer for this important comment. We agree with the reviewer that the sequence covered by the peptide HR13 from fibronectin type II domain of FXII has been shown by Citarella et al. to be involved in binding negatively charged surfaces²⁴. Additionally, HR13 shares sequence similarity in 8 amino acids out of 13 with the previously published peptide YHK9 which blocks FXII binding to HUVECs (Mahdi et al.)²⁵. We want to pinpoint that these identified residues in FN2 domain were based on our AlphaFold2 simulations results which were in accordance with the previously published work by Citarella et al. and Mahdi et al. To address this comment, we included the citations for HR13 in our main text and in the supplementary table. In the results section we write in the revised version:

“The area covered by the peptide HR13 derived from the FN2 domain of FXII has been previously shown to mediate binding to negatively charged surfaces⁴⁴. Furthermore, HR13 shares sequence similarity in 8 amino acids out of 13 with the previously published peptide YHK9 which blocks FXII binding to HUVEC cells².”

(page 7, line 11-14)

Comment 1-18:

p. 9, lines 24,25. Need to say that a local source, i.e., PTCs themselves when activated could be the source of the cofactor zinc ion.

Response 1-18:

We thank the reviewer for this comment. Zinc transporters have been shown to be expressed by kidney tubular cells, mediating reabsorption of zinc from urinary filtrate followed by zinc storage in tubular cells³². Furthermore, intracellular zinc homeostasis is maintained by the heavy metal binding family of metallothioneins³³. Metallothionein expression is induced in tubular cells exposed to high glucose and deficiency of metallothionein exacerbates DKD in mice by inducing oxidative stress^{34,35}. Collectively, these data imply that tubular injury in DKD could promote the release of zinc from its intracellular stores, which will enhance FXII binding to uPAR and hence FXII signaling. This point is addressed in the revised discussion:

“Alternatively, renal tubular cells express zinc transporters and reabsorb zinc⁷⁸. Additionally, deficiency of metallothioneins, a family of heavy metal binding proteins expressed by tubular cells and maintaining zinc homeostasis, exacerbates DKD in murine models by inducing oxidative stress^{79,80}. Therefore, it appears possible that tubular injury in hyperglycemic conditions releases zinc from its intracellular stores, enhancing FXII binding to uPAR and hence FXII signaling.”

(page 12, line 5-9)

Comment 1-19:

p.9, line 29. Refs #35 and 36 before Ref 18 focused on examining the FXII zymogen and FXIIa issue in the FXII/FXIIa interaction with cells.

Response 1-19:

We thank the reviewer to bringing this point to our attention. We have rearranged the citations based on the reviewer's suggestion as the following:

“Our data suggest that the effects of FXII on tubular cells are independent of its activation, in line with previous reports on other cell types²⁻⁴.”

(page 12, line 14-16)

Comment 1-20:

Fig 8b. This experiment shows that after diabetes, a morpholino for FXII knockdown in mice resulted in reduced albumin urinary loss at 24 weeks over 16 weeks. I understand how FXII could influence PTCs' activity via the uPAR-beta-1-signaling system, how does FXII control albumin levels in urine? Do proximal tubules influence albumin loss? If not, how do the authors explain this interesting result.

Response 1-20:

We thank the reviewer for this insightful comment. Indeed, PTCs reabsorb albumin filtered by glomeruli under physiological conditions. Albumin will appear in urine only if the reabsorption ability of PTCs is overwhelmed or hampered due to tubular damage^{36,37}. Furthermore, PTCs play an important role in maintaining the tubuloglomerular feedback, hence the tubular injury associated with DKD causes impairment in this feedback mechanism contributing to hemodynamic changes and hyperfiltration leading to albuminuria³⁸. This point is addressed now in the revised discussion as the following:

“Based on the current results we assume that the protective phenotype observed with FXII morpholino treatment as reflected by reduced tubular senescence and ameliorated albuminuria reflects the contribution of tubular cell injury to albuminuria in DKD⁷⁴.”

(page 11, line 29-31)

Reviewer #2 (Remarks to the Author):

The authors identifies the role of zymogen FXII signaling in the progression of DKD suggesting that FXII binds to uPAR and signals via integrin β 1 on tubular cells, promoting DNA damage and senescence. They also indicate that therapies targeting FXII or the FXII-uPAR interaction constitute new strategies to ameliorate DKD. It is a well-designed and interesting study. I suggest acceptance of the paper after the following concerns being resolved.

We were pleased by the reviewer's positive and constructive comments. The points raised by the reviewer are addressed as follows:

Comment 2-1:

The authors suggested that zymogen FXII induced DNA damage and senescence in tubular cells, whereas the involved mechanism is unknown.

Response 2-1:

We appreciate this comment by the reviewer. In our work we focused on the receptor mechanism. As such, we demonstrate that FXII binding to uPAR induces uPAR-integrin β 1 signaling, promoting ROS generation, DNA damage, and senescence in tubular cells. Additionally, the RNA-seq data revealed downregulation of pathways related to integrin signaling, focal adhesions and Rho GTPase signaling in hyperglycemic *F12*^{-/-} mice compared to hyperglycemic WT mice kidneys (**Fig. R2-1a**). Persistent integrin β 1 signaling is known to mediate cellular adhesion through phosphorylation of focal adhesion kinase (FAK), and the latter acts as a scaffolding platform for other kinases including Src kinase^{39,40}. Constitutive integrin signaling associated with abnormal focal adhesions is linked to senescence through increased ROS production⁴¹. Furthermore, Src maintains a senescence phenotype in fibroblasts in response to DNA damage⁴². Integrin signaling activates the Rho family GTPase Rac1 which is promoted by FAK^{10,40,43}. In turn, Rac1 is known to activate NADPH oxidases (NOXs) to produce ROS causing DNA damage and senescence^{44,45}.

Following the reviewer's suggestion and based on these published insights, we exposed HKC-8 cells to FXII (62 nM) in the presence of zinc (10 μ M). FXII time-dependently increased the phosphorylation of FAK and Src and upregulated the expression of Rac1 and NOX1 (**Fig. R2-1b,c**). To determine whether this pathway is activated upon FXII-uPAR interaction we pretreated cells with the inhibitory uPAR based peptides DL19 and DV20 (300 μ M) or a functional blocking monoclonal antibody targeting integrin β 1 (10 μ g/ml). Both interventions abolished FXII's effects on focal adhesion kinases activation or the upregulation of Rac1 and NOX1 (**Fig. R2-1d-g**).

Taken together, the newly generated data identify a signaling mechanism through which persistent uPAR-integrin β 1 signaling axis activates focal adhesion kinases and the downstream Rac1 leading to NOX1 activation and increased ROS production promoting DNA damage and senescence. This point is currently addressed in the results section:

“Aberrant integrin β 1 signaling is associated with abnormal focal adhesions contributing to senescence⁵². To investigate whether signaling of FXII-uPAR-integrin β 1 axis modulates focal adhesions, we determined phosphorylation of focal adhesion kinase (FAK) and Src kinase in HKC-8 cells exposed to FXII. FXII time-dependently induced phosphorylation and activation of FAK and Src, which was associated with upregulation of the Rho family GTPase Rac1 and of the ROS regulator NADPH oxidase 1 (NOX1) (Fig. S20a,b). To determine whether this pathway is activated

upon FXII-uPAR interaction we pretreated cells with the inhibitory uPAR-based peptides DL19 and DV20 (300 μ M) or a functional blocking monoclonal antibody targeting integrin β 1 (10 μ g/ml). Both interventions abolished FXII's effects on focal adhesion kinases activation or the upregulation of Rac1 and NOX1 (Fig.S20c-f). Collectively, these data support a model in which the integrin α 6 β 1 heterodimer mediates FXII-uPAR intracellular signaling via FAK-Src, thereby promoting DNA damage and senescence in kidney tubular cells."

(page 8, line 43- page 9, line 6)

and in the discussion:

"Integrin signaling activates the Rho family GTPase Rac1 which is promoted by FAK⁹⁴⁻⁹⁶. Rac1 activates NADPH oxidases (NOXs), increasing ROS generation, DNA damage and senescence^{97,98}. Constitutive integrin signaling associated with abnormal focal adhesions induces senescence through increased ROS production⁵². Integrin β 1 signaling mediates cellular adhesion through phosphorylation of FAK, and the latter acts as a scaffolding platform for other kinases including Src kinase^{95,99}. Src activation maintains a senescence phenotype in fibroblasts in response to DNA damage¹⁰⁰. In the current study, FXII deficiency was associated with downregulation of pathways related to integrin signaling, focal adhesions and Rho GTPase signaling in hyperglycemic kidneys. Furthermore, interference with FXII-uPAR binding or blocking integrin β 1 reduced activation of focal adhesion kinases and upregulation of Rac1 and NOX1 in tubular cells, suggesting the involvement of abnormal focal adhesions in FXII-mediated oxidative DNA damage."

(page 12, line 37- page 13, line 2)

Fig. R2-1 (a) Bar graph representing the top enriched pathways based on the downregulated differentially expressed genes (DEGs) in *F12*^{-/-}-DM compared to WT-DM kidneys using KEGG (Kyoto Encyclopedia of Genes and Genomes), WikiPathways, Reactome, PID (Pathway Interaction Database), and GO (Gene Ontology: Biological processes) databases. The pathways were ranked by the false discovery rate (FDR). (b,c) Representative immunoblots (b, loading control: α -Tubulin) and dot plots summarizing results (c) for p-FAK, total FAK p-Src, total Src, Rac1 and NOX1 expression in HKC-8 cells exposed to purified human FXII (62 nM) in the presence of Zn²⁺ (10 μ M) for 6, 24, and 48 h. Dot plots reflecting mean \pm SEM of 3 independent experiments; one-way ANOVA with Tukey's multiple comparison test. *P<0.05, **P<0.01, ***P<0.001. (d,e) Representative immunoblots (d, loading control: α -Tubulin) and dot plots summarizing results (e) for p-FAK, total FAK p-Src, total Src, Rac1 and NOX1 expression in HKC-8 cells exposed to purified human FXII (62 nM) in the presence of Zn²⁺ (10 μ M) for 24 h with and without pretreatment with peptides DL19 and DV20 (300 μ M) for 1 h compared to control non-treated cells. Dot plots reflecting mean \pm SEM of 3 independent experiments; one-way ANOVA with Tukey's multiple comparison test. **P<0.01, ***P<0.001. (f,g) Representative immunoblots (f, loading control: α -Tubulin) and dot plots summarizing results (g) for p-FAK, total FAK p-Src, total Src, Rac1 and NOX1 expression in HKC-8 cells exposed to purified human FXII (62 nM) in the presence of Zn²⁺ (10 μ M) for 24 h with and without pretreatment with a monoclonal integrin β 1 function blocking antibody (10 μ g/ml) for 30 min. compared to control non-treated cells. Dot plots reflecting mean \pm SEM of 3 independent experiments; one-way ANOVA with Tukey's multiple comparison test. **P<0.01, ***P<0.001.

Comment 2-2:

The coagulation pathway and inflammatory effects of FXII which also contribute to progression of DKD might be checked.

Response 2-2:

We thank the reviewer for this comment. The protease FXII initiates the intrinsic coagulation pathway by cleavage of coagulation factor XI and inflammation through cleavage of prekallikrein to liberate bradykinin⁴⁶. We concur with the reviewer that coagulation and inflammatory pathways initiated by FXII may contribute to the progression of DKD. Based on reviewer's suggestion and in order to investigate whether there is a change in coagulation parameters between hyperglycemic WT and *F12*^{-/-} mice, we determined plasma levels of D-Dimer using a commercially available ELISA. We did not find differences between both genotypes (**Fig. R2-2a**).

Furthermore, transcriptomic analysis of the Nephroseq[®] and the Karokidney public databases revealed increased kidney tubular *F12* expression in DKD patients compared to controls, while the expression of other contact pathway genes including *KLKB1* (encoding kallikrein), *KNG1* (encoding high molecular weight kininogen), and *F11* (encoding FXI) were downregulated in DKD patients (**Fig. R2-2b,c**). We confirmed downregulation of *Klkb1* and *Kng1* in kidneys of hyperglycemic mice using qPCR and did not observe differences between WT and *F12*^{-/-} mice (**Fig. R2-2d**). Interestingly, expression of *F11* mRNA was already down-regulated in control *F12*^{-/-} mice compared to control WT mice, indicating an unknown interaction of *F11* and *F12* in the kidney. Importantly, in hyperglycemic WT and *F12*^{-/-} mice, the mRNA levels of *F11* were comparable. This observation requires future independent investigations.

Regarding the activation of the kallikrein-kinin-system (KKS) by the protease FXII and in the context of DKD, controversial results have been reported in the past. A number of studies showed a protective effect of exogenous kallikrein administration in DKD models, and that deficiency of bradykinin receptors aggravates the kidney phenotype and enhance senescence in diabetic mice⁴⁷⁻⁴⁹. On the other hand, other studies reported that bradykinin promotes tubular and glomerular injury^{50,51}. Bradykinin receptors (BR1 and BR2) are required for the normal kidney function as they regulate nitric oxide production and hence the GFR⁵². Of note, *F12*^{-/-} mice show only a 50% reduction of bradykinin levels, as bradykinin can be liberated through other mechanisms including the action of prolylcarboxypeptidase^{53,54}. Hence, basal levels of bradykinin required for normal kidney function may be maintained in *F12*^{-/-} mice. Considering these controversial data, it is difficult to address the role of FXII in regard to KKS regulation in DKD.

In preliminary analyses we did not observe different expression of bradykinin receptors (*Bdkrb1* and *Bdkrb2*) in the kidneys of WT and *F12*^{-/-} mice by qPCR (data not shown). Further experimental approaches are needed to fully address the role of bradykinin-receptor signaling in the current experimental model. Given the controversial data, dissecting the role of FXII on bradykinin signaling in DKD may be challenging and beyond the scope of the current manuscript.

Finally, we want to emphasize, that we used two FXII cyclic inhibitor peptides (FXII-618 and FXII-900) and the enzymatic dead mutant FXII-Locarno in our original manuscript. Both approaches demonstrate that markers of DNA damage and senescence induced by FXII were not affected by these interventions (**Fig. R2-2f-k**), illustrating that the observed phenotype is independent of the enzymatic activity and hence not linked to coagulation or KSS activation.

Taken together, the newly generated data in addition to the data that we have shown in our original manuscript indicates that the observed effects of FXII are zymogen functions and are not related, at least in part in our context, to its enzymatic activity. This point is addressed now in the results section:

“Markers of DNA damage and senescence were not affected by the cyclic inhibitors or the expression of the Locarno mutant (Fig. S14). Furthermore, transcriptomic analysis of the Nephroseq® and the Karokidney public databases revealed that in addition to *KNG1* (Fig. S11a,b), the expression of the contact pathway genes *KLKB1* (encoding kallikrein) and *F11* (encoding FXI) were downregulated in DKD patients (Fig. S15a,b). We confirmed downregulation of *Klkb1* and *Kng1* in hyperglycemic mice kidneys using qPCR and did not find a difference between WT and *F12*^{-/-} mice (Fig. S15c). Furthermore, in hyperglycemic WT and *F12*^{-/-} mice, mRNA levels of *F11* were comparable, as were D-dimer plasma levels, reflecting that coagulation activation was not different between genotypes (Fig. S15c,d).”

(page 8, line 1-8)

and in the discussion:

“Our data suggest that the effects of FXII on tubular cells are independent of its activation, in line with previous reports on other cell types²⁻⁴. Furthermore, the differential regulation of the contact pathway proteins in the kidneys of DKD patients and in hyperglycemic mice and the absence of coagulation changes in *F12*^{-/-} mice compared to WT mice support the notion that the effect of FXII is independent of coagulation activation.”

(page 12, line 14-18)

Fig. R2-2 (a) dot plot summarizing plasma levels of D-Dimer in normoglycemic controls (C) and hyperglycemic (DM) wild type (WT) and $F12^{-/-}$ mice. Dot plot reflecting mean \pm SEM of 5 mice per group; two-way ANOVA with Tukey's multiple comparison test. ns: non-significant. (b,c) Heatmaps summarizing *KLKB1*, *KNG1*, *F11*, and *F12* expression in the tubulointerstitial compartment comparing diabetic kidney disease patients (DKD) to non-diabetic control healthy donors (C) in Karokidney (b) and Nephroseq (c) public databases. Unpaired student's t-test. (d) Dot plots representing the expression of *Klkb1*, *Kng1* and *F11* expression in experimental groups (as described in a, qPCR). Dot plots reflecting mean \pm SEM of 4 mice per group; unpaired student's t-test, ** P <0.01, *** P <0.001, ns: non-significant

Comment 2-3:

The SASP should also be analyzed by ELISA besides q-pcr.

Response 2-3:

We appreciate this suggestion by the reviewer. We agree with the reviewer that data supporting regulation of SASP markers on the protein level will be supportive. Based on the reviewer's suggestion, we analyzed kidney lysates of WT and *F12^{-/-}* mice using a Olink® Target 48 Cytokine Panel which contains a number of cytokines, chemokines, and growth factors related to the SASP such as Il-4, Il-5, Il-22, Cxcl11, Fgf21, and Csf1. The analysis revealed that these SASP-associated factors tended to be positively correlated with albuminuria across both genotypes (hyperglycemic WT and *F12^{-/-}* mice, correlation was not significant, probably due to the low sample size and heterogenous data in the WT-DM group). Of note, while at least some SASP-associated factors were increased WT-DM mice, but not in *F12^{-/-}* DM mice. These data support that the SASP phenotype is associated with kidney injury and tends to be stronger in hyperglycemic WT mice (**Fig. R2-3**).

To address this, we state in the revised manuscript:

“Furthermore, we analyzed a panel of genes known to be associated with senescence in mice^{35,36}. These genes were downregulated in the kidneys of hyperglycemic *F12^{-/-}* compared to WT mice (Fig. 4d). The downregulation of genes related to cell cycle arrest, the senescence associated secretory phenotype (SASP), cell adhesion, and fibrosis in the kidneys of hyperglycemic *F12^{-/-}* mice was confirmed by qRT-PCR (Fig. 4e). Additionally, analysis of selected SASP-related cytokines and chemokines using Olink technology revealed increased SASP-related cytokines and chemokines in some hyperglycemic WT, but not in hyperglycemic *F12^{-/-}* mice kidneys (Fig. S5d).”

(page 5, line 27-33)

Fig. R2-2 Scatter plots representing the expression of selected SASP-related factors in kidney lysates in relation to albuminuria levels in hyperglycemic WT and *F12^{-/-}* mice. The SASP factors were detected using Olink technology.

References (only related to response letter):

1. Hayek, S.S., *et al.* A tripartite complex of suPAR, APOL1 risk variants and alpha(v)beta(3) integrin on podocytes mediates chronic kidney disease. *Nature medicine* **23**, 945-953 (2017).
2. Zhu, K., *et al.* The D2D3 form of uPAR acts as an immunotoxin and may cause diabetes and kidney disease. *Science translational medicine* **15**, eabq6492 (2023).
3. Matlin, K.S., Haus, B. & Zuk, A. Integrins in epithelial cell polarity: using antibodies to analyze adhesive function and morphogenesis. *Methods* **30**, 235-246 (2003).
4. Mezu-Ndubuisi, O.J. & Maheshwari, A. The role of integrins in inflammation and angiogenesis. *Pediatric research* **89**, 1619-1626 (2021).
5. Pang, X., *et al.* Targeting integrin pathways: mechanisms and advances in therapy. *Signal transduction and targeted therapy* **8**, 1 (2023).
6. Zhang, G. & Eddy, A.A. Urokinase and its receptors in chronic kidney disease. *Frontiers in bioscience : a journal and virtual library* **13**, 5462-5478 (2008).
7. Wu, C.Z., *et al.* Urokinase plasminogen activator receptor and its soluble form in common biopsy-proven kidney diseases and in staging of diabetic nephropathy. *Clinical biochemistry* **48**, 1324-1329 (2015).
8. Lieberthal, W., *et al.* Beta1 integrin-mediated adhesion between renal tubular cells after anoxic injury. *Journal of the American Society of Nephrology : JASN* **8**, 175-183 (1997).
9. Zhou, X., *et al.* An integrin antagonist (MK-0429) decreases proteinuria and renal fibrosis in the ZSF1 rat diabetic nephropathy model. *Pharmacology research & perspectives* **5**(2017).
10. Jun, J.I. & Lau, L.F. The matricellular protein CCN1 induces fibroblast senescence and restricts fibrosis in cutaneous wound healing. *Nature cell biology* **12**, 676-685 (2010).
11. Schmaier, A.H. & Stavrou, E.X. Factor XII - What's important but not commonly thought about. *Research and practice in thrombosis and haemostasis* **3**, 599-606 (2019).
12. Kirita, Y., Wu, H., Uchimura, K., Wilson, P.C. & Humphreys, B.D. Cell profiling of mouse acute kidney injury reveals conserved cellular responses to injury. *Proceedings of the National Academy of Sciences of the United States of America* **117**, 15874-15883 (2020).
13. Wu, H., Kirita, Y., Donnelly, E.L. & Humphreys, B.D. Advantages of Single-Nucleus over Single-Cell RNA Sequencing of Adult Kidney: Rare Cell Types and Novel Cell States Revealed in Fibrosis. *Journal of the American Society of Nephrology : JASN* **30**, 23-32 (2019).
14. Megyesi, J., Andrade, L., Vieira, J.M., Jr., Safirstein, R.L. & Price, P.M. Positive effect of the induction of p21WAF1/CIP1 on the course of ischemic acute renal failure. *Kidney international* **60**, 2164-2172 (2001).
15. Megyesi, J., *et al.* Increased expression of p21WAF1/CIP1 in kidney proximal tubules mediates fibrosis. *American journal of physiology. Renal physiology* **308**, F122-130 (2015).
16. Al-Dabet, M.M., *et al.* Reversal of the renal hyperglycemic memory in diabetic kidney disease by targeting sustained tubular p21 expression. *Nature communications* **13**, 5062 (2022).
17. LaRusch, G.A., *et al.* Factor XII stimulates ERK1/2 and Akt through uPAR, integrins, and the EGFR to initiate angiogenesis. *Blood* **115**, 5111-5120 (2010).
18. Mahdi, F., *et al.* Mapping the interaction between high molecular mass kininogen and the urokinase plasminogen activator receptor. *The Journal of biological chemistry* **279**, 16621-16628 (2004).
19. Bdeir, K., *et al.* The kringle stabilizes urokinase binding to the urokinase receptor. *Blood* **102**, 3600-3608 (2003).
20. Ploug, M. Identification of specific sites involved in ligand binding by photoaffinity labeling of the receptor for the urokinase-type plasminogen activator. Residues located at equivalent positions in uPAR domains I and III participate in the assembly of a composite ligand-binding site. *Biochemistry* **37**, 16494-16505 (1998).

21. Hu, L., *et al.* Why Senescent Cells Are Resistant to Apoptosis: An Insight for Senolytic Development. *Frontiers in cell and developmental biology* **10**, 822816 (2022).
22. Soto-Gamez, A., Quax, W.J. & Demaria, M. Regulation of Survival Networks in Senescent Cells: From Mechanisms to Interventions. *Journal of molecular biology* **431**, 2629-2643 (2019).
23. Rea, I.M., *et al.* Age and Age-Related Diseases: Role of Inflammation Triggers and Cytokines. *Frontiers in immunology* **9**, 586 (2018).
24. Citarella, F., te Velthuis, H., Helmer-Citterich, M. & Hack, C.E. Identification of a putative binding site for negatively charged surfaces in the fibronectin type II domain of human factor XII—an immunochemical and homology modeling approach. *Thrombosis and haemostasis* **84**, 1057-1065 (2000).
25. Mahdi, F., Madar, Z.S., Figueroa, C.D. & Schmaier, A.H. Factor XII interacts with the multiprotein assembly of urokinase plasminogen activator receptor, gC1qR, and cytokeratin 1 on endothelial cell membranes. *Blood* **99**, 3585-3596 (2002).
26. Citarella, F., *et al.* Structure/function analysis of human factor XII using recombinant deletion mutants. Evidence for an additional region involved in the binding to negatively charged surfaces. *European journal of biochemistry* **238**, 240-249 (1996).
27. Ravon, D.M., Citarella, F., Lubbers, Y.T., Pascucci, B. & Hack, C.E. Monoclonal antibody F1 binds to the kringle domain of factor XII and induces enhanced susceptibility for cleavage by kallikrein. *Blood* **86**, 4134-4143 (1995).
28. Tian, Z. & Liang, M. Renal metabolism and hypertension. *Nature communications* **12**, 963 (2021).
29. Forbes, J.M. & Thorburn, D.R. Mitochondrial dysfunction in diabetic kidney disease. *Nature reviews. Nephrology* **14**, 291-312 (2018).
30. Hudgins, A.D., *et al.* Age- and Tissue-Specific Expression of Senescence Biomarkers in Mice. *Frontiers in genetics* **9**, 59 (2018).
31. Knoppert, S.N., Valentijn, F.A., Nguyen, T.Q., Goldschmeding, R. & Falke, L.L. Cellular Senescence and the Kidney: Potential Therapeutic Targets and Tools. *Frontiers in pharmacology* **10**, 770 (2019).
32. Ranaldi, G., Perozzi, G., Truong-Tran, A., Zalewski, P. & Murgia, C. Intracellular distribution of labile Zn(II) and zinc transporter expression in kidney and MDCK cells. *American journal of physiology. Renal physiology* **283**, F1365-1375 (2002).
33. Chen, B., *et al.* Cellular zinc metabolism and zinc signaling: from biological functions to diseases and therapeutic targets. *Signal transduction and targeted therapy* **9**, 6 (2024).
34. Ogawa, D., *et al.* High glucose increases metallothionein expression in renal proximal tubular epithelial cells. *Experimental diabetes research* **2011**, 534872 (2011).
35. Tachibana, H., *et al.* Metallothionein deficiency exacerbates diabetic nephropathy in streptozotocin-induced diabetic mice. *American journal of physiology. Renal physiology* **306**, F105-115 (2014).
36. Comper, W.D., Haraldsson, B. & Deen, W.M. Resolved: normal glomeruli filter nephrotic levels of albumin. *Journal of the American Society of Nephrology : JASN* **19**, 427-432 (2008).
37. Eriguchi, M., *et al.* Renal tubular ACE-mediated tubular injury is the major contributor to microalbuminuria in early diabetic nephropathy. *American journal of physiology. Renal physiology* **314**, F531-F542 (2018).
38. Tonneijck, L., *et al.* Glomerular Hyperfiltration in Diabetes: Mechanisms, Clinical Significance, and Treatment. *Journal of the American Society of Nephrology : JASN* **28**, 1023-1039 (2017).
39. Schlaepfer, D.D. & Mitra, S.K. Multiple connections link FAK to cell motility and invasion. *Current opinion in genetics & development* **14**, 92-101 (2004).
40. Choma, D.P., Milano, V., Pumiglia, K.M. & DiPersio, C.M. Integrin alpha3beta1-dependent activation of FAK/Src regulates Rac1-mediated keratinocyte polarization on laminin-5. *The Journal of investigative dermatology* **127**, 31-40 (2007).

41. Shin, E.Y., *et al.* Integrin-mediated adhesions in regulation of cellular senescence. *Science advances* **6**, eaay3909 (2020).
42. Anerillas, C., *et al.* Early SRC activation skews cell fate from apoptosis to senescence. *Science advances* **8**, eabm0756 (2022).
43. Chang, F., Lemmon, C.A., Park, D. & Romer, L.H. FAK potentiates Rac1 activation and localization to matrix adhesion sites: a role for betaPIX. *Molecular biology of the cell* **18**, 253-264 (2007).
44. Salazar, G. NADPH Oxidases and Mitochondria in Vascular Senescence. *International journal of molecular sciences* **19**(2018).
45. Shi, Y., *et al.* Rac1-Mediated DNA Damage and Inflammation Promote Nf2 Tumorigenesis but Also Limit Cell-Cycle Progression. *Developmental cell* **39**, 452-465 (2016).
46. Long, A.T., Kenne, E., Jung, R., Fuchs, T.A. & Renne, T. Contact system revisited: an interface between inflammation, coagulation, and innate immunity. *Journal of thrombosis and haemostasis : JTH* **14**, 427-437 (2016).
47. Liu, W., *et al.* Exogenous kallikrein protects against diabetic nephropathy. *Kidney international* **90**, 1023-1036 (2016).
48. Tomita, H., Sanford, R.B., Smithies, O. & Kakoki, M. The kallikrein-kinin system in diabetic nephropathy. *Kidney international* **81**, 733-744 (2012).
49. Pincon-Raymond, M., *et al.* Conditional immortalization of normal and dysgenic mouse muscle cells by the SV40 large T antigen under the vimentin promoter control. *Developmental biology* **148**, 517-528 (1991).
50. Tang, S.C., *et al.* Bradykinin and high glucose promote renal tubular inflammation. *Nephrology, dialysis, transplantation : official publication of the European Dialysis and Transplant Association - European Renal Association* **25**, 698-710 (2010).
51. Tan, Y., Wang, B., Keum, J.S. & Jaffa, A.A. Mechanisms through which bradykinin promotes glomerular injury in diabetes. *American journal of physiology. Renal physiology* **288**, F483-492 (2005).
52. Kakoki, M., McGarrah, R.W., Kim, H.S. & Smithies, O. Bradykinin B1 and B2 receptors both have protective roles in renal ischemia/reperfusion injury. *Proceedings of the National Academy of Sciences of the United States of America* **104**, 7576-7581 (2007).
53. Iwaki, T. & Castellino, F.J. Plasma levels of bradykinin are suppressed in factor XII-deficient mice. *Thrombosis and haemostasis* **95**, 1003-1010 (2006).
54. Zhu, L., *et al.* Role of prolylcarboxypeptidase in angiotensin II type 2 receptor-mediated bradykinin release in mouse coronary artery endothelial cells. *Hypertension* **56**, 384-390 (2010).

REVIEWERS' COMMENTS

Reviewer #1 (Remarks to the Author):

Overview. This paper is an outstanding work. It builds upon prior knowledge on FXII-cell interactions and opens it up to the field of nephrology.

I have no further recommendations at this time.

Reviewer #2 (Remarks to the Author):

The authors have addressed all the comments and suggestions. I suggested acceptance.